# Neural alignment predicts learning outcomes in students taking an introduction to computer science course

Meir Meshulam [1,2✉], Liat Hasenfratz[1,2], Hanna Hillman [1,2], Yun-Fei Liu[1,2], Mai Nguyen[1,2], Kenneth A. Norman [1,2] & Uri Hasson [1,2]

Despite major advances in measuring human brain activity during and after educational experiences, it is unclear how learners internalize new content, especially in real-life and online settings. In this work, we introduce a neural approach to predicting and assessing learning outcomes in a real-life setting. Our approach hinges on the idea that successful learning involves forming the right set of neural representations, which are captured in canonical activity patterns shared across individuals. Specifically, we hypothesized that learning is mirrored in neural alignment: the degree to which an individual learner's neural representations match those of experts, as well as those of other learners. We tested this hypothesis in a longitudinal functional MRI study that regularly scanned college students enrolled in an introduction to computer science course. We additionally scanned graduate student experts in computer science. We show that alignment among students successfully predicts overall performance in a final exam. Furthermore, within individual students, we find better learning outcomes for concepts that evoke better alignment with experts and with other students, revealing neural patterns associated with specific learned concepts in individuals.

[1] Princeton Neuroscience Institute, Princeton University, Princeton, NJ, USA. [2] Department of Psychology, Princeton University, Princeton, NJ, USA.
✉email: meshulam@princeton.edu

Learning plays a central role in shaping our cognition. As we gain new knowledge, our thinking changes: as physicist Richard Feynman observed, "The world looks so different after learning science"[1]. Recently, multivariate "brain reading" analysis techniques have significantly advanced our understanding of how knowledge is represented in neuronal activity[2–4]. These methods, together with representational similarity analysis, have made it possible to delineate the fine-grained structure of neural representations of learned knowledge, and to link neural patterns to specific knowledge across multiple domains[2,5–9]. For the most part, this body of work has examined well-established concept representations (e.g. representations of objects and animals[10–12]) rather than newly acquired concepts.

Recent imaging work has begun addressing this issue, extending a large body of work that has studied changes in neuronal circuits during and after learning[13,14]. The current state of the art in assessing acquired conceptual knowledge based on neural data is exemplified by Cetron et al.[15]. In that study, engineering students and novices were presented with photos of real-world structures and asked to consider the forces acting on them. Neural classifiers were then trained to predict an expert-defined category label (cantilevers/trusses/vertical loads) for each item based on imaging data. These enabled the authors to detect individual differences in understanding the physics concept of Newtonian forces. In an earlier study, Mason and Just[16] reported a progression of activation throughout the cortex during learning, providing group-level "snapshots" of the various cortical networks activated as participants progressed through explanations about four simple mechanical systems.

By design, these studies were conducted under carefully controlled experimental conditions. They employed a small set of stimuli and categories within a narrow context which allowed researchers to use custom-built, domain-specific classifiers. However, to bridge the gap to real-world education we need to develop general-purpose methods that can be applied to neural data from an actual course (covering multiple topics) and are capable of pinpointing which of those topics have (and have not) been successfully learned. Unlike in the controlled setup that has benefited earlier studies, in a typical college course students are required to communicate with instructors and peers; actively use a variety of static and dynamic learning resources inside and outside of class; assimilate multiple new concepts simultaneously; integrate course material over a prolonged period of several weeks; and often master new skills. Therefore, we aimed to study learning in a real-life setting: a "flipped" introduction to computer science course in which students watched lecture videos outside of class. The broad range of topics covered by the course enabled us to look beyond overall performance measures and examine learning at high resolution, i.e., specific topics in individual students.

Key to our approach is how new information is communicated to students and integrated in memory during learning. Communication between individuals has been linked to neural coupling, such that (i) the brains of speakers and listeners show joint response patterns, and (ii) more extensive speaker–listener neural coupling enables better communication[17–19]. Likewise, when people watch the same video, shared activity patterns emerge across the brain[20]. Recent imaging studies have shown that memories of this shared experience are encoded in a similar way across individuals, particularly in default mode network (DMN) regions[21,22]. Furthermore, a study that compared neural activity time courses in children and adults during short educational videos found that the degree to which children showed adult-like brain responses during math videos was correlated with their math test scores[23]. Notably, specific concepts have also been shown to evoke similar neural activity patterns across individuals,

suggesting a shared structure for neural representations[2,24–27]. This body of work suggests that shared neural responses reflect thinking alike. In the context of learning, the students' goal could be viewed as laying the neural foundation that would allow them to think like experts.

Here, we focused on science, technology, engineering, and mathematics (STEM) learning in academia. Our goal was to use shared neural activity patterns across learners and experts to quantify and predict learning outcomes in a popular course at Princeton University. We tested the hypothesis that learning is mirrored in neural alignment: the degree to which individual learners' neural representations match canonical representations observed in experts. Our findings demonstrate that alignment during video lectures over the course of a semester successfully predicted final exam performance. Critically, we also obtained fine-grained, concept-specific signatures of understanding in individual brains. While verbally answering open exam questions, alignment between students and experts and alignment between students and classmates in medial cortical regions were both positively correlated with performance across questions, within individual students. Furthermore, a consistent set of relationships between topics emerged across students, correlating with performance within individual students and revealing how different concepts were integrated together. These results reveal neural activity patterns that reflect successful learning of specific topics in individual participants within a broad-ranging, real-world STEM course.

## Results

Did alignment to canonical neural representations emerge during learning, and did alignment reflect successful learning? To address these questions, we examined neural activity patterns and learning outcomes in undergraduate students and in graduate experts. In collaboration with the Department of Computer Science at Princeton University, we recruited undergraduate students enrolled in COS 126: Computer Science—An Interdisciplinary Approach. The course introduces basic concepts in programming and computer science using a flipped classroom model, with lecture videos watched outside of class. Students underwent functional magnetic resonance imaging (fMRI) scans five times during a 13-week semester while watching a subset of that week's video lectures in the scanner. Subjects were asked not to view these lecture videos online before the scans. The subset of lectures shown in each scan was approximately 40 min long and comprised 3–5 segments (21 segments, 197 min in total). On the final week of the semester, students were shown—in the scanner —five 3-min lecture recap videos with the highlights from previous weeks, followed by a final exam (Fig. 1a and Table 1).

To establish a baseline, the same exam was also given to students at the beginning of the semester, in written form ("pre" exam). Graduate experts underwent the final scan only, watching the recap videos from all lectures and completing the final exam. The exam was self-paced, with exam questions (16 in total) spanning a variety of course topics from programming to theory (see Supplementary Text for exam questions). In the final exam, participants were asked to give verbal responses to visually presented questions (mean response length 31.9 s, s.d. 24.7). Questions were scored individually by course staff, providing a fine-grained measure of understanding. All students received a score of zero on the baseline exam (Fig. 1b). This confirmed that students had no prior knowledge of course material. By the end of the course, all students demonstrated knowledge gains (pre-post comparison, two-sided $t$-test, $t(19) = -12.6$, $p < 0.001$), with substantial variance across students (range 22–76 out of 100, median 53.1, s.d. 17.1).

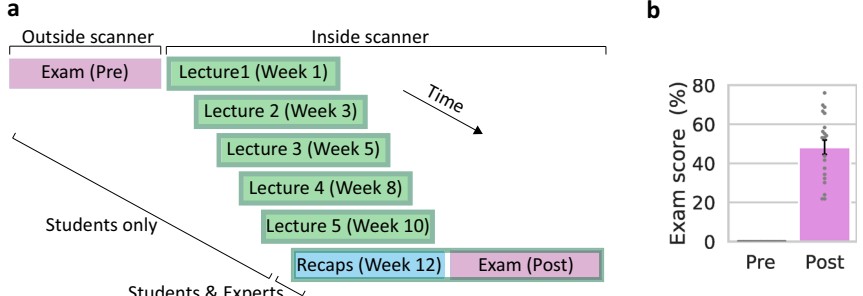

**Fig. 1 Study design and exam scores. a** Study design. Students enrolled in an introduction to computer science course underwent six fMRI scans throughout the course. During the first five scans, students were shown course lecture videos. On the final scan (bottom), students were shown lecture recaps and given a final exam. Experts underwent the final scan only. See Table 1 for stimuli and task details. **b** Exam scores. Pretest (left) was performed prior to scanning, posttest (right) was performed during scan 6. Individual students ($n = 20$) are shown in gray. Error bar, ±1 s.e.m.

**Table 1 Stimuli and tasks.**

| Stimulus | Participants | Task | Length of time bin for neural pattern | Total stimulus length |
|---|---|---|---|---|
| Lecture videos | Students | Passive viewing | 30 s of video (fixed) | 197 min |
| Recap videos | Students+Experts | Passive viewing | 30 s of video (fixed) | 16 min |
| Exam (in scanner) | Students+Experts | Verbal response | Entire question (variable) | 10–22 min |

**Prediction of learning outcomes from neural activity during lectures.** Our first goal was to predict learning outcomes from brain activity during lecture videos. To this end, we calculated neural alignment-to-class across all lectures, comparing each student's response patterns to the mean response patterns across all other students (Fig. 2a and "Methods"). Alignment values varied across the cortex, with the strongest values recorded in visual occipital regions, auditory and language regions, and parts of the default mode and attention networks (Fig. 2b). The alignment map was qualitatively in line with the body of literature showing that watching the same video elicits shared activity patterns across individuals[20,28]. However, in the current work, alignment maps were not thresholded (i.e. statistical analysis for alignment effects was not performed), and all voxels were included in the subsequent searchlight analysis in which we correlated alignment and exam scores. This was done in order to test whether variance in alignment was related to variance in scores and avoid excluding brain regions (e.g. the hippocampus) where alignment values were lower yet could be predictive of learning outcomes. Correlation between alignment and exam scores was done using a between-participants design, first in eight anatomically defined regions of interest (ROIs) and then across the entire cerebral cortex using a searchlight analysis. Our ROIs included major nodes of the DMN and the hippocampus as well as control regions in early sensory cortex and the amygdala. Our selection of ROIs was motivated by findings that activity in the DMN during memory encoding of new content (real-life stories or audiovisual movies) predicted recall success for that material[21,22,29]. The searchlight analysis enabled us to look for regions showing a correlation between alignment and learning outcomes in a data-driven manner. Throughout the manuscript, searchlight size was $5 \times 5 \times 5$ voxels ($15 \times 15 \times 15$ mm cubes), and statistical significance evaluated using a one-sided permutation test (creating null distributions by shuffling labels 1000 times), controlling the false discovery rate (FDR) to correct for multiple comparisons at $q = 0.05$ (refs. [12,30]).

Alignment-to-class in ROIs during lecture videos showed a significant positive correlation with final exam scores in the angular gyrus, precuneus, anterior cingulate cortex (ACC) (all overlap with the DMN), and the hippocampus, as well as early visual and auditory areas (Fig. 2c, d and Table 2). Across ROIs, the highest correlation values were observed in the hippocampus, allowing the most reliable prediction of learning outcomes. Alternative measures of alignment-to-class, in which we varied the length of the time bin used for calculating spatial alignment, yielded similar though somewhat weaker results (Supplementary Table 1); likewise, we obtained similar results using a measure of shared responses in the time domain (temporal Inter-Subject Correlation, ISC; see Supplementary Table 1 for results using these measures). Our cortical searchlight analysis showed multiple brain regions where students' alignment-to-class predicted their final exam scores (Fig. 2d). In line with the ROI analysis results, these regions included anterior and posterior medial (PM) areas as well as the bilateral angular gyrus, key nodes of the DMN. In addition, we observed significant correlations in temporal and insular cortex. A power analysis revealed that prediction improved as more data were aggregated across lectures (Supplementary Fig. 1 and Supplementary Results).

**Neural alignment between students and experts.** Neural alignment-to-class was strongly correlated with alignment-to-experts. Experts were scanned during recap videos (16 min in total) and while taking the final exam. We separately calculated alignment-to-class and alignment-to-experts for each student in each task and then correlated these measures using a between-participants design (see "Methods"). The goal of this analysis was to examine whether alignment-to-class reflected convergence on expert patterns. Figure 3a shows results in an example ROI in ACC during recaps, while Fig. 3b shows results in the same ROI during the exam. In both tasks, alignment-to-class and alignment-to-experts were positively correlated across all ROIs (Table 3). A searchlight analysis revealed that these effects extended to large parts of cortex, including the default mode and attention networks. Cortical maps for recaps and the exam are shown in Fig. 3c, d respectively. These results indicate that the mean responses across all students converge to the average, or canonical, responses seen in experts during both recaps and the

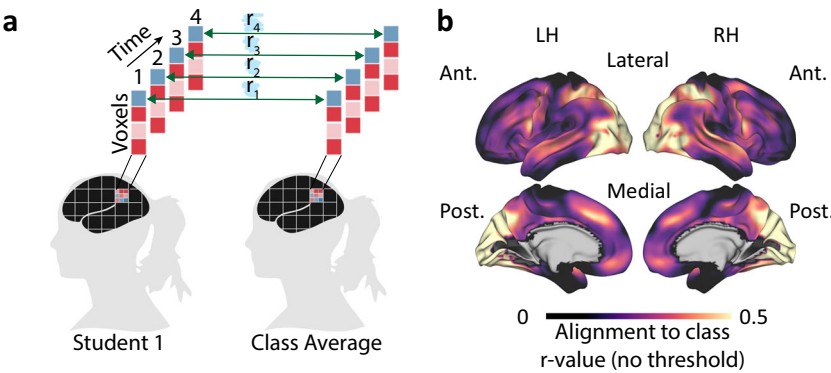

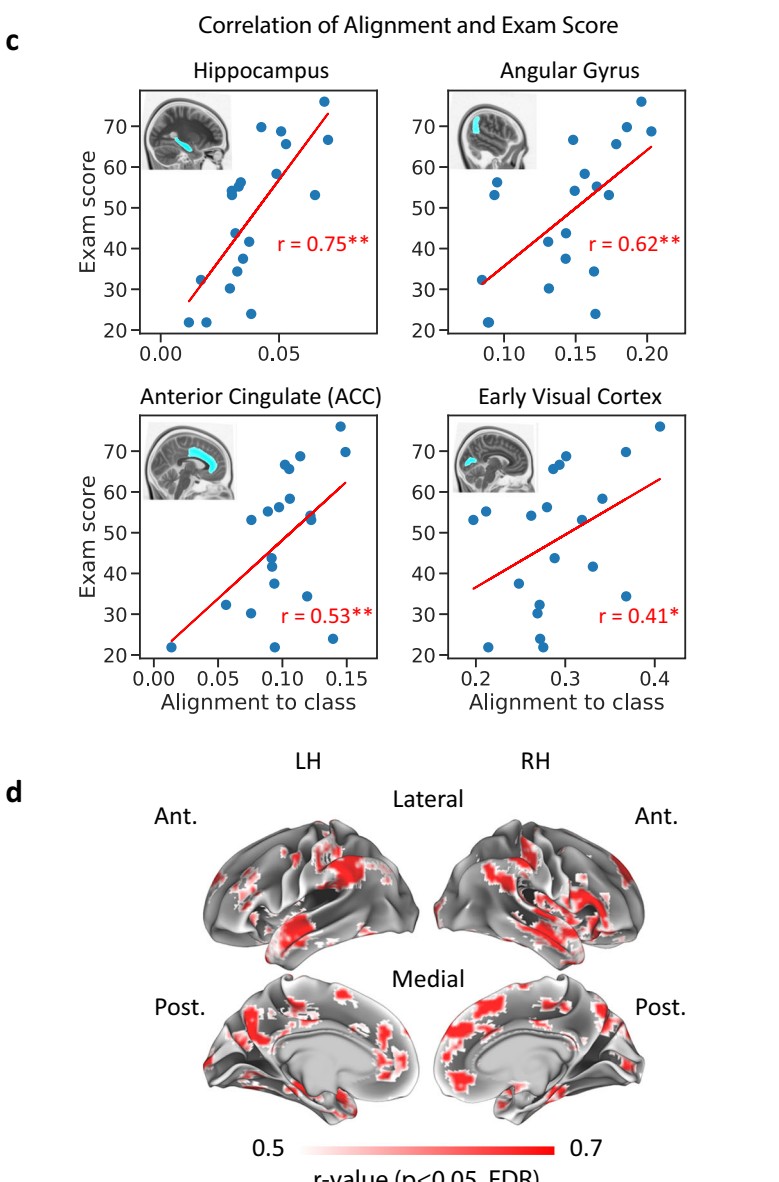

final exam. Furthermore, it indicates that the individual differences seen across subjects in their ability to be aligned to *class* are preserved when we look at their ability to converge to canonical *expert* responses. Next, we asked whether the ability of each student to converge to these canonical responses predicted learning outcomes during the final exam.

**Think like an expert: assessing learning success during exam using expert canonical responses**. We hypothesized that better alignment to experts and to peers during question answering would be linked to better answers. To test this hypothesis, we obtained spatial activity patterns during each question and calculated same-question alignment-to-experts and alignment-to-class

**Fig. 2 Alignment-to-class during lectures predicts final exam scores. a** Calculation of alignment-to-class during lecture videos. **b** Alignment-to-class across the entire cerebral cortex. For demonstration purposes, this non-thresholded map shows mean alignment-to-class values across time and across students. **c** Prediction of exam scores from neural alignment in example ROIs. Mean alignment-to-class across lectures (x-axis) is correlated with exam scores (y-axis) using a between-participant design. Blue dots represent individual students. Asterisks denote significant correlation (one-sided permutation test, corrected), *p < 0.05, **p < 0.01. See Table 2 for a summary of ROI results. **d** Prediction of exam scores from neural alignment across the cortex. Searchlight analysis results shown. Voxels showing significant correlation are shown in color (one-sided permutation test, p < 0.05, corrected). LH left hemisphere, RH right hemisphere, Ant. anterior, Post. posterior.

**Table 2 Prediction of exam scores from neural alignment in ROIs.**

| Region of interest (ROI) | Lectures | Exam | | | |
| | | "Same question" | | "Knowledge structure" | |
| | Corr. exam score and alignment-to-class | Corr. exam score and alignment-to-class | Corr. exam score and alignment-to-experts | Corr. exam score and alignment-to-class | Corr. exam score and alignment-to-experts |
|---|---|---|---|---|---|
| Angular gyrus | 0.62** | 0.14* | 0.10 (n.s.) | 0.17* | −0.03 (n.s.) |
| Ant. cingulate (ACC) | 0.53** | 0.28** | 0.23** | 0.21** | 0.00 (n.s.) |
| Hippocampus | 0.75** | 0.16* | −0.06 (n.s.) | 0.13* | −0.09 (n.s.) |
| Post. sup. temporal gyrus | 0.40* | 0.25** | 0.17* | 0.23** | 0.05 (n.s.) |
| Precuneus | 0.61** | 0.10 (n.s.) | 0.18* | 0.03 (n.s.) | −0.06 (n.s.) |
| *Amygdala* | 0.29 (n.s.) | 0.06 (n.s.) | −0.03 (n.s.) | −0.04 (n.s.) | 0.07 (n.s.) |
| *Early auditory* | 0.46* | 0.07 (n.s.) | −0.02 (n.s.) | 0.01 (n.s.) | 0.08 (n.s.) |
| *Early visual* | 0.41* | 0.11 (n.s.) | 0.12 (n.s.) | 0.05 (n.s.) | −0.02 (n.s.) |

Correlation between spatial alignment measures and exam score during lectures and during the final exam. Results are shown in DMN ROIs as well as in control regions (text in italics) in sensory cortex (visual, intracalcarine cortex; auditory, Heschl's gyrus) and subcortex (amygdala). See Supplementary Table 1 for correlation results obtained using alternative ways to calculate alignment-to-class. Asterisks denote significant correlation (one-sided permutation test, corrected across ROIs). *p < 0.05, **p < 0.01, n.s. not significant.

scores (Fig. 4a, see "Methods" for details). These scores allowed us to quantify how the neural patterns evoked by each question were related to the neural patterns evoked by the same question in other participants. We correlated alignment and question scores separately (across questions) within each student (see Fig. 4b for an example from a single student in a single ROI), and then took the mean across all students (Fig. 4c). Importantly, this within-participant design allowed us to capitalize on between-questions variability while controlling for individual differences.

Alignment-to-experts and alignment-to-class were both positively correlated with exam scores across several ROIs. In the ACC and superior temporal ROIs, alignment-to-class and alignment-to-experts were both positively correlated with exam scores (Fig. 4d and Table 2). Exam scores were also significantly correlated with alignment-to-experts in the precuneus and alignment-to-class in the hippocampus, angular gyrus, and visual ROIs. Our searchlight analysis results supported these findings, highlighting regions across anterior and PM cortex bilaterally (Fig. 4d). Importantly, both alignment-to-experts and alignment-to-class searchlight results highlighted these medial cortical regions. These findings show that neural alignment of specific question-by-question patterns was associated with better learning outcomes, indicating that concepts that were represented more similarly to the experts (and the class) were the concepts that students better understood. The results further highlight the ACC and medial prefrontal (mPFC) regions as areas where both alignment-to-class and alignment-to-experts were significantly correlated with behavior.

We then turned to examine the link between neural alignment and behavior while controlling for response length. This was motivated by the possibility that (i) longer answers might have yielded more stable spatial patterns, and that (ii) response length and quality could be linked (e.g. better answers could be longer). Therefore, a possible alternative explanation for our results is that they were driven by response length. To address this, we used a within-participant regression model to predict question scores

from answer length. This model yielded a residual error term for each question ("residual score", predicted score minus true score). We then repeated our original analysis using the residual score instead of the true score for each question. This procedure yielded cortical maps that were highly similar to those shown here (Supplementary Fig. 2 a, b). Thus, across all brain areas showing a link between alignment and exam performance, effects were robust to response length.

**Knowledge structure reflects learning in individual students.** In the next set of analyses, we asked how individual concepts were integrated together in learners' brains. Specifically, we hypothesized that learning new concepts also entails learning their contextual relations to other concepts. For example, the concepts "binary tree" and "linked list" are related in a specific way (a linked list can be used to implement a binary tree). To test this, we first created a "knowledge structure" for each participant, capturing the set of relationships between neural patterns evoked by different questions. For each question in each participant, we measured the similarity of the neural pattern evoked by that question to the canonical patterns evoked by *other* questions (in the class average or in the experts). We defined the set of question-specific relationships as the knowledge structure for that question for that participant. To predict performance, we then compared that question-specific knowledge structure (for that participant) to the question-specific knowledge structure for the experts (alignment-to-experts) or for the class as a whole (alignment-to-class) (Fig. 5a). The resulting alignment scores were then correlated with question scores using a within-participant design (Fig. 5b).

We found that knowledge structure alignment was positively correlated with exam scores across the hippocampus, ACC, angular gyrus and temporal ROIs when derived for the student cohort (alignment-to-class, Table 2). In line with this, our searchlight analysis showed robust results for alignment-to-class,

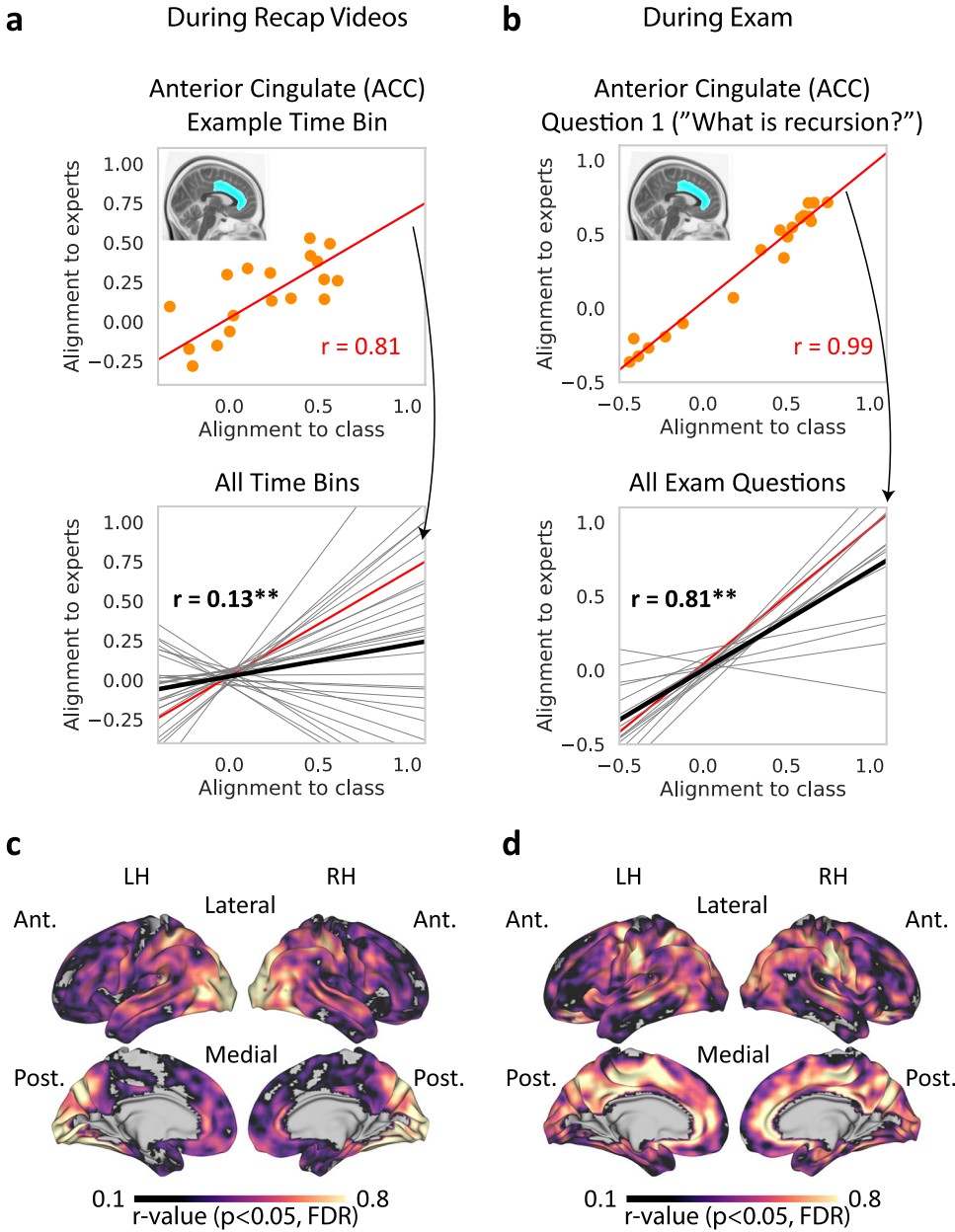

**Fig. 3 Alignment-to-class and alignment-to-experts are positively correlated across the brain.** Correlation between alignment-to-class and alignment-to-expert during recap videos (left) and during final exam (right) are shown. **a** Between-subjects correlation during recap videos, in a single ROI. Top, correlation in a single example 30-s time bin. Orange dots represent individual students. Bottom, mean across all time bins (solid black line). Trendlines for individual time bins are shown in gray, with the example time bin shown in red. Asterisks denote significant correlation (one-sided permutation test, corrected), $*p < 0.05$, $**p < 0.01$. **b** Between-subjects correlation during the final exam, in a single ROI. Top, correlation during the first question. Orange dots represent individual students. Bottom, mean across all exam questions (solid black line). Trendlines for individual questions are shown in gray, with the example question shown in red. Asterisks denote significant correlation (one-sided permutation test, $p < 0.01$, corrected). See Table 3 for a summary of ROI analysis results. **c** Correlation during recap videos across the cortex, searchlight analysis results are shown. Voxels showing significant correlation are shown in color (one-sided permutation test, $p < 0.05$, corrected). **d** Correlation during the final exam across the cortex. Voxels showing significant correlation are shown in color (one-sided permutation test, $p < 0.05$, corrected). LH left hemisphere, RH right hemisphere, Ant. anterior, Post. posterior.

highlighting medial cortical regions (Fig. 5c). Furthermore, the searchlight analysis showed a remarkable correspondence between knowledge structure and same-question results, with both maps highlighting similar medial regions (Figs. 4d and 5c). ROI and searchlight analysis results for alignment-to-experts were not significant across all regions ($p > 0.05$, corrected). While alignment-to-experts searchlight results were qualitatively similar

to alignment-to-class results (Supplementary Fig. 3, $p < 0.01$, uncorrected), no voxels survived multiple comparisons correction. In sum, these results showed that (i) students' exam performance was significantly tied to their ability to create—and reinstate—a specific set of relationships between neural representations; and that (ii) the anatomical regions involved in knowledge structure alignment showed high correspondence with

**Table 3 Alignment-to-experts is positively correlated with alignment-to-class during recaps and during final exam.**

| Region of interest (ROI) | Corr. alignment-to-class and alignment-to-experts | |
|---|---|---|
| | Recap videos | Exam |
| Angular gyrus | 0.47** | 0.49** |
| Ant. cingulate (ACC) | 0.13** | 0.81** |
| Hippocampus | 0.08* | 0.47** |
| Post. sup. temporal gyrus | 0.25** | 0.49** |
| Precuneus | 0.43** | 0.67** |
| *Amygdala* | 0.09* | 0.37** |
| *Early auditory* | 0.12** | 0.74** |
| *Early visual* | 0.63** | 0.49** |

Correlation between alignment-to-class and alignment-to-experts is shown during lectures and during the exam. Results are shown in DMN ROIs as well as in control regions (text in italics) in sensory cortex (visual, intracalcarine cortex; auditory, Heschl's gyrus) and in subcortex (amygdala). Asterisks denote significant correlation (one-sided permutation test, corrected across ROIs). *$p < 0.05$, **$p < 0.01$, n.s. not significant.

regions involved in same-question alignment. As a control, we repeated this analysis while controlling for response length, again obtaining highly similar results (Supplementary Fig. 2c).

**Effects in DMN regions across tasks**. Across our dataset, we repeatedly observed a link between learning outcomes and neural alignment in mPFC regions, PM regions, left angular gyrus, and medial temporal gyrus. We therefore performed an intersection analysis to substantiate this observation and determine whether the same, or different, voxels in these regions emerged across tasks. This analysis highlighted voxel clusters in anterior medial cortex, as well as in PM cortex and superior temporal cortex, that showed significant effects across all alignment-to-class analyses (Fig. 6a). This set of regions overlaps in large part with the DMN. Furthermore, the intersection of the correlation map of same-question alignment-to-experts with exam scores and the correlation map of same-question alignment-to-class with exam scores yielded a similar map (Fig. 6b). These results indicated a key role for DMN regions across different phases of learning and further emphasized the link between alignment-to-experts and alignment-to-class measures.

**Discussion**
In this work, we introduce a neural approach to predicting and assessing learning outcomes in a real-life setting. This approach hinges on the idea that successful learning involves forming the right neural representations, which are captured in canonical activity patterns shared across learners and experts. In the current study, we put forward the notion that learning is mirrored in neural alignment: the degree to which individual learners' neural representations match canonical representations observed in experts. We tested this hypothesis in students enrolled in an introduction to computer science course and in graduate student experts, using a longitudinal fMRI design. Our findings show that across regions involved in memory encoding and reinstatement in the DMN and hippocampus, alignment successfully predicted overall student performance in a final exam. Furthermore, within individual students, learning outcomes were better for concepts that evoked better alignment. We discuss the role of neural alignment in learning and understanding below.

**Neural alignment successfully predicts learning outcomes**.
During learning, neural activity patterns in each student participant comprised both common and idiosyncratic components. This is in line with a growing body of work showing that neural

alignment across individuals watching the same video or listening to the same audio narrative is positively correlated with the level of shared context-dependent understanding[21,22,25,27]. Our results show that alignment-to-class was strongly correlated with exam score: across students, stronger similarity to the class predicted better performance (Fig. 2 and Table 2). This observation held across the DMN, implicated in internally focused thought and memory[31–33]. These results dovetail with findings that better alignment to common patterns in these regions support better memory for shared experiences[21,22], as well as with recent electroencephalogram findings linking higher temporal synchrony (inter-subject correlation, ISC) during short educational videos with higher motivation and better learning outcomes[34,35].

Our finding that similarity to the class during lectures predicted performance is also in line with imaging results reported by Cantlon and Li[23]. In that study, the authors used temporal ISC to show that children's scores in a standardized math test were correlated with the degree to which they showed adult-like brain responses during educational math videos. These correlations were localized to the intraparietal sulcus, a region previously implicated in numerical processing. We observed similar effects in our temporal ISC analysis as well as in our analysis of spatial patterns (Fig. 2 and Supplementary Fig. 1). The current study further extended the existing body of work to a real-world college course setting, enabling us to directly assess learning outcomes of course-specific material from brain activity and demonstrate a role for the DMN in learning, discussed below.

A key point here is that, in our flipped class, a significant part of learning occurred outside of lectures. Given the structure of the course, it is unlikely that alignment-to-class during any specific lecture directly reflected learning success of lecture topics at the end of the course. The first viewing of a course lecture, like the first reading of a textbook chapter, is just the beginning of a learning process that includes repetition and practice. Furthermore, only a fraction of course lecture segments (~1/7 of total) was shown to student participants in the scanner, while performance was measured using an exam deliberately designed to span the entire course. Therefore, to explain the predictive power of alignment-to-class, we need to go beyond lecture-specific effects. We submit that neural alignment to common patterns reflects the online, moment-to-moment process of learning within individuals. Furthermore, the results indicate that monitoring such a process can predict to some extent the outcome of the learning process. This claim is supported by the finding that it is possible to reliably predict learning outcomes from neural activity during the early weeks of the course (Supplementary Fig. 1).

**Class patterns reflect expert patterns**. To understand why alignment with the class leads to improved performance, we need to consider what shared class patterns may reflect. One possibility is that these patterns reflect group knowledge. According to this view, when individual patterns are averaged and idiosyncratic differences cancel out, what emerges is a good approximation of an ideal canonical representation. We would like to suggest this is analogous to a "central limit theorem" of knowledge: the mean is a reflection of the fact that most students, most of the time, follow the lecture as intended: what they share is the correct interpretation of course material. Similar ideas have been conceptualized as the wisdom of crowds. On the other hand, in a class where the norm is to struggle with the material, rather than understand it well, common response patterns may not emerge. Another caveat is that common misunderstandings would also be reflected in the common pattern. These misunderstandings, however, would not be shared by experts, resulting in high

Alignment During Exam

alignment across students but low alignment between students and experts.

We hypothesized that canonical class patterns, reflecting successful learning, would match expert patterns. We tested this hypothesis during recaps (short summaries of the lecture videos, shown just prior to the exam) and during the exam. Our findings

confirmed that alignment-to-class and alignment-to-experts were positively correlated across large swaths of the cerebral cortex, including in DMN regions (Fig. 3). The tight link between alignment-to-class and alignment-to-experts suggests that students and experts may converge on a single set of shared neural states. However, we observed substantial variability in correlation

**Fig. 4 Same-question alignment during the exam correlates with performance. a** Left, student and class patterns are correlated on a question-by-question basis to derive alignment-to-class during exam. Right, student and expert patterns are similarly correlated to derive alignment-to-experts. **b** Within-subject correlation between alignment and exam score in a single ROI, in a single student. Violet dots represent individual exam questions. Left, correlation between alignment-to-class and exam score. Right, correlation between alignment-to-experts and exam score. **c** Within-subject correlation between alignment and exam score in a single ROI, trendlines for all students shown. Red, the trendline of the student shown in panel **b**. Black, mean across all students. Left, correlation between alignment-to-class and exam score. Right, correlation between alignment-to-experts and exam score. Asterisks denote significant correlation (one-sided permutation test, corrected), *$p < 0.05$, **$p < 0.01$. For all ROI analysis results, see Table 2. **d** Correlation across the cortex, searchlight analysis results shown. Voxels showing significant correlation are shown in color (one-sided permutation test, $p < 0.05$, corrected). Left, correlation between alignment-to-class and exam score. Right, correlation between alignment-to-experts and exam score. Control analyses for response length are shown in Supplementary Fig. 2a and 2b. Note the correspondence between the two maps in major DMN nodes on the medial surface. LH left hemisphere, RH right hemisphere, Ant. anterior, Post. posterior.

magnitude between these two alignment measures across cortical regions and between tasks (Table 3). We speculate that these differences could be linked to the different nature of the tasks (passive viewing of recap videos and active generation of verbal responses during the exam). Nevertheless, the factors contributing to this variability remain unknown at this point.

**Alignment tracks understanding of specific topics**. In direct support of our hypothesis, we found that alignment and learning outcomes were correlated on a fine-grained, question-by-question basis, within individual students. Our results show that during the exam, alignment-to-experts and alignment-to-class were both positively correlated with question scores (Fig. 4). Importantly, these results were specific to the neural patterns observed in the experts for each particular question and thus were robust to individual differences. In other words, our results could not emerge due to some students being better learners than others, or having better working memory, for example. The effects tested here could only emerge if, in individual students, answers that evoked better alignment-to-experts obtained higher scores and vice versa.

To our knowledge, this is the first demonstration of shared structure in neural responses across individuals during open question answering. Formulating an answer required participants to call upon their memory and understanding of question-specific concepts, as well as more general cognitive processes such as language production. Importantly, exam questions were highly complex and required integrative answers, aimed at assessing conceptual understanding and making it difficult to account for students' performance based on subsequent memory effects or rote recall alone. The correlation of alignment and performance emerged most strongly in medial DMN regions, suggesting that the aligned neural patterns in these areas supported introspection and memory (Fig. 6b). It is therefore possible that successful alignment reflected understanding, particularly in light of the body of work linking similarity in DMN regions to better understanding of narratives[21,22]. This opens up the future possibility of using alignment to assess learning success, offering a different perspective than traditional performance measures.

**Learning the right "knowledge structure"**. The more abstract a concept, the less it is grounded in physical reality. This has posed a challenge to teachers, who need to build a structure of interrelated ideas from the ground up. In a course like Introduction to Computer Science, the understanding of basic concepts (e.g. algorithms) later facilitates the introduction of more advanced theoretical concepts (e.g. intractability). The alignment of knowledge structures across students provides a fine-grained measure of success in learning-specific topics. It shows how each topic is grounded in others, revealing the interaction between mental representations. This result could therefore allow

examining understanding in individual learners in high resolution. While same-question alignment-to-class could show, for example, that the concept of intractability was not well understood, knowledge structure alignment could show that the underlying reason is difficulty with the more basic concept of recursion.

Together with our same-question results, the positive correlation between knowledge-structure alignment and exam performance provides key support for our hypothesis (Fig. 5 and Table 2). The correlation between alignment-to-class during lectures and exam performance, discussed above, could be explained in terms of coarse-grained individual-difference variables, even if there is no direct link between alignment-to-class and learning outcomes (e.g., conscientious students may show high alignment-to-class during lectures because they closely attend to videos, and obtain good exam scores because they study a lot outside of class, leading to a correlation between alignment-to-class and exam performance). Crucially, the analyses relating same-question alignment and knowledge-structure alignment to performance on specific exam questions were conducted within-subjects, and thus the results of these analyses cannot be explained in terms of differences between participants.

Our results further show that, even at the end of the course, students' degree of knowledge structure convergence to experts did not predict students' question-answering accuracy. A likely cause of this null result is lack of power (fewer experts than students in our dataset). However, an alternative explanation is that alignment to novice peers is actually a better measure of learning than alignment to experts, even when power is matched between these measures. This could also account for the stronger student-to-class effects found in our same-question analysis (Table 2 and Fig. 4d). One reason for this could be that experts' grasp of course material is more holistic, drawing on their broader understanding of the field, and therefore qualitatively different from that of students. In this case, the class average could provide a unique window into the neural state best associated with successful learning. Resolving this issue would require conducting power-matched comparisons between students and experts. However, if this hypothesis is substantiated, it would call for a reconsideration of the traditional focus on expert-like thinking in expertise research in favor of group-like thinking among learners.

In sum, these results show that the set of relationships between mental representations of abstract concepts is behaviorally relevant, and point to medial DMN regions as key nodes supporting these representations. A promising direction for future work lies in leveraging recent methodological advances (e.g., in language modeling and network science methods) to track the development of students' knowledge structure during learning[36–38]. These methods could make it possible to delineate the exact relationship between student and expert knowledge structures in the DMN.

Knowledge Structure Alignment During Exam

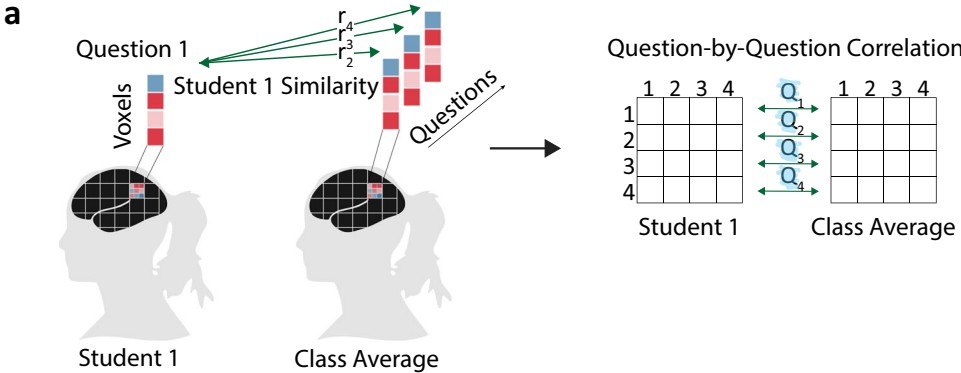

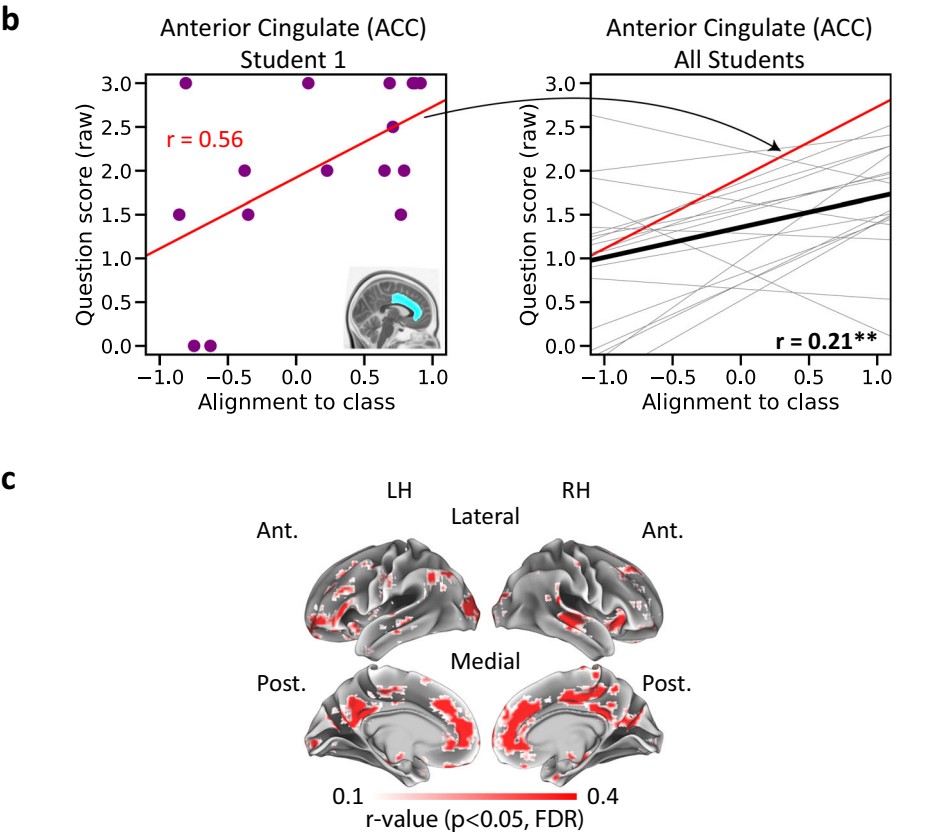

**Fig. 5 Knowledge structure alignment during the exam correlates with performance. a** Student and class knowledge structures are correlated on a question-by-question basis to derive knowledge structure alignment during the exam. Cell i, j in the student's knowledge structure is the correlation between the student's pattern for question i with the class pattern for question j. Student and mean class knowledge structures are then correlated on a row-by-row (question-by-question) basis. **b** Left, within-subject correlation between alignment-to-class and exam score in a single ROI, in a single student. Each violet dot represents a single question. Right, within-subject correlation between alignment-to-class and exam score in a single ROI, trendlines for all students shown. Red, the trendline of the student shown on the left. Black, mean across all students. Asterisks denote significant correlation (one-sided permutation test, corrected), *$p < 0.05$, **$p < 0.01$. **c** Correlation between knowledge structure alignment-to-class and exam score, searchlight analysis results shown. Voxels showing significant correlation are shown in color (one-sided permutation test, $p < 0.05$, corrected). A control analysis for response length is shown in Supplementary Fig. 2c. Searchlight results for alignment-to-experts are shown in Supplementary Fig. 3. Note the correspondence between the alignment-to-class maps here and in Fig. 4. LH left hemisphere, RH right hemisphere, Ant. anterior, Post. posterior.

**A key role for medial DMN regions during learning**. The significance of DMN cortical structures in our results highlights the role these regions play during key stages of the learning process, from first exposure to course video lectures to review of learned material (recap videos) and, finally, question answering during the exam (Fig. 6). Effects in these regions were robust, emerging

for both alignment measures (to class and to experts). These findings are in line with previous work that localized behaviorally relevant, memory-related shared representations to these areas[21,22]. They are also consistent with earlier findings that specific patterns of activity during memory encoding in DMN regions predicted recall performance[29], as well as with a recent

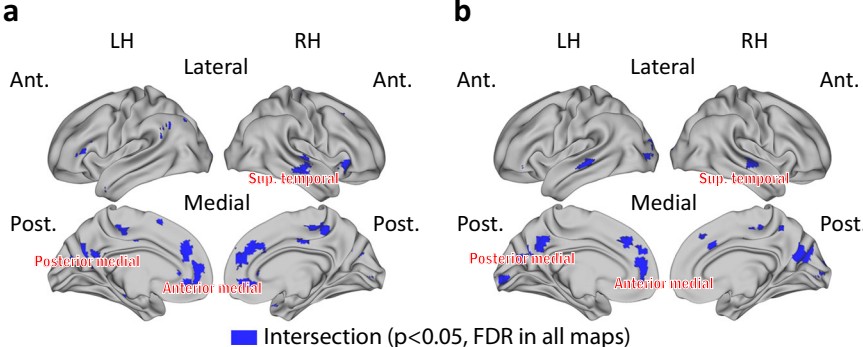

**Fig. 6 Robust neural alignment effects in medial and temporal cortical regions emerge across all analyses. a** Overlap regions across all three datasets and analyses for alignment-to-class. Blue color indicates voxels in the intersection set of the following maps: (i) correlation between alignment-to-class during lectures and exam scores (shown in Fig. 2d), (ii) correlation between alignment-to-class and alignment-to-experts during recaps (shown in Fig. 3c), (iii) correlation between same-question alignment-to-class during the final exam and exam score (shown in Fig. 4d, left panel), and (iv) correlation between knowledge structure alignment-to-class during the exam and exam score (shown in Fig. 5c). **b** Overlap regions for same-question analyses, blue color indicates voxels in the intersection set of the following maps: (i) correlation between same-question alignment-to-class during the final exam and exam score (shown in Fig. 4d, left panel) and (ii) correlation between same-question alignment-to-experts during the final exam and exam score (shown in Fig. 4d, right panel).

report of hippocampal changes triggered by learning the structures and names of organic compounds[39]. The PM cortical system plays a key role in episodic memory[40] as well as forming part of the DMN. However, studies that examined the neuronal correlates of math and science have generally highlighted cortical areas outside the DMN. For example, parietal and frontal regions have been shown to play a key role in mathematical cognition[41,42]. In physics, concepts such as gravity and frequency have each been associated with a distinctive set of cortical regions, mostly on the lateral cortical surface[24], and recent work using multivariate methods has localized representations of physics concepts to dorsal fronto-parietal regions and ventral visual areas[15]. One way to account for these apparent discrepancies and for the prominence of DMN regions in our outcome-based results is to consider the likely role of different cortical regions in learning. While specific types of cognitive operations may well be subserved by distinct sets of cortical regions, long-term learning requires forming the correct neural representations, encoding them in memory and retrieving them in the right context, all hallmarks of the DMN and its associated structures.

An unexpected finding is that, during lectures, we also observed correlations between exam scores and alignment-to-class in early sensory areas (Fig. 2 and Table 2). A possible explanation for such correlations during lectures is that these correlations reflected top-down effects that interacted with the way students processed visual and auditory information (for example, it is possible that stronger learners attended to specific details in the lectures video which were missed by less attentive students).

**Limitations**. Our results were derived from scanning a cohort of students enrolled in a single course at a single campus. Further research is required in order to ensure that they generalize to other domains and learning settings. Although we see no a priori reason why our findings should be limited to any particular type of course (e.g. courses in STEM, or introductory courses), or a particular type of college, further research is required to rule out these possibilities. Another limitation concerns the relatively small number of participants: our longitudinal design allowed us to collect a large amount of data (>3 h of functional scans) per student participant, but limited the number of participants that we could run given available scanning resources. Although the number of student participants in the current study is in line with

previous studies from our group[21,43,44], it may have limited our ability to detect smaller effects. To determine how much lecture data were required to obtain robust correlations with exam scores, we conducted a power analysis (Supplementary Results and Supplementary Fig. 1) that simulated a situation where scanning was halted after a single lecture segment, two segments, and so forth. Importantly, while some cortical regions in the DMN showed a robust correlation between alignment-to-class and exam scores given relatively small amounts of data, other cortical regions only showed a robust correlation given the entire dataset. It is possible that even more regions would have shown significant effects if we had tested more participants or collected more data per participant. The silver lining of this result is that simpler and shorter experimental designs than those employed here may be sufficient to study real-world learning effects in DMN regions. Finally, like other neural measures, alignment between individuals likely reflects multiple contributing factors. These could include similarities in students' educational background and familiarity with the teaching method. Mapping the different factors and their unique contributions remains a topic for future work.

## Methods

**Participants and stimuli**. Twenty-four "student" and five "expert" participants (11 females) were recruited for the study. All participants were right handed, had normal or corrected-to-normal vision and hearing, and reported no learning disabilities. All except one expert were native English speakers. Student participants reported having no prior knowledge or experience in computer science. Prior to scanning, all students completed the course placement exam (described under "Stimuli") in written form and received 0 out of 3 points on all questions (see grading details below). Experts all had an undergraduate or graduate degree in computer science and reported having significant programming experience as well as knowledge of introduction to computer science course material (≥6 on a 7-point Likert scale). Participants received monetary compensation for their time. The study complied with all relevant ethical regulations for work with human participants and informed consent was obtained in accordance with experimental procedures approved by the Institutional Review Board at Princeton University.

Students were enrolled in COS 126: Computer Science—An Interdisciplinary Approach (lectures available at informit.com/title/9780134493831) at Princeton University and were taking the course for the first time. The course sets out to teach basic principles of computer science in the context of scientific, engineering, and commercial applications. It uses a "flipped" classroom model, with students viewing lecture videos on their own schedule and interacting with course staff in precepts and class meetings. All students took the course for credit and participated in the course normally, with the exception that they were asked to view part of the lecture videos (~3 h out of ~21 h in total) in the scanner. Participants were asked not to view these lecture videos online before the scans. Students were scanned every 2–3 weeks during a single semester (Fig. 1). Four students dropped the course

and were excluded from the experiment. Two student datasets were incomplete (one student skipped scan 3; one student's exam scan data was not collected due to experimenter error). One expert did not complete the exam. The final sample consisted of 20 datasets collected from undergraduate students (18 complete) and five expert datasets (four complete). No statistical methods were used to predetermine sample sizes. Our student sample size is similar to those reported in previous publications from our group[21,43,44].

Stimuli included video lectures, recaps, and a final exam, shown throughout a series of six scans. During each of the first five scans, students watched 3–5 segments of course lecture videos that were required viewing for the following week (mean segment length 9 min, total of ~40 min shown in each scan, total of 21 segments in all scans). At the end of each scan, students were given a set of questions about the lecture (question data not analyzed in the current manuscript). In addition, on scans 3–5, students were shown two 3-min recap videos, summarizing the previous two lectures shown (these data were not analyzed in the current manuscript). On the final scan, students watched all five 3-min recap videos, each summarizing a single lecture ("recaps"). This was followed by an exam that required verbal responses. The same stimuli were shown to all students. Experts underwent the final scan only. Each lecture segment and recap was shown in a separate scanner run. At the beginning and end of each run, we appended 20–30 s of unrelated "filler" audiovisual clips (from YouTube "oddly satisfying" compilations, featuring, for example, objects being assembled neatly). Filler clips were similarly added in previous studies from our group[21,25]. This was done because the stimulus onset may elicit a global arousal response, which could add noise to the analysis. To avoid this, scan data collected during fillers, as well as during the first 12 s of each video, were omitted from analysis. Exam stimuli consisted of 16 written questions, shown in fixed order. We used the course placement exam, developed by course staff for the benefit of students wishing to demonstrate proficiency in course material without taking it. Questions were designed to span the breadth of material covered in the course; some required distilling large concepts into simple explanations and others were more practical. Exam scores were not used by course staff to assess students' performance. The same exam was used to assess students' knowledge prior to scanning (in written form) and on the final scan (with verbal responses).

**Experimental procedures.** Participants were asked to watch lecture videos as they normally would. Lecture videos were shown at normal speed (first scan) or slightly accelerated (1.15× speed, scans 2–5, accelerated for all participants at their request). Stimuli were projected using an LCD projector onto a rear-projection screen located in the scanner bore and viewed with an angled mirror. PsychoPy2 was used to display the stimuli and synchronize them with MRI data acquisition[45]. Audio was delivered via in-ear headphones (Sensimetrics S14), and the volume was adjusted for every participant before each scan. Video monitoring was used to monitor participants' alertness in about 40% of scans at random (Eyelink, SR Research). Monitoring showed that no participants fell asleep during the experiment. Verbal responses to exam questions were recorded using a customized MR-compatible recording system (FOMRI III, OptoAcoustics). Motion during speech was minimized by instructing participants to remain still and by stabilizing participants' heads with foam padding, as in previous studies from our group[22,25,46]. Participants indicated end-of-answer using a handheld response box (Current Designs).

No outside resources were available during the videos or the exam, and students could not take notes. No feedback was provided to participants during the exam. The exam was self-paced with a "Please Wait" text slide presented for 12 s between questions. Question text was shown for the entire length of the answer at the center of the screen, and participants confirmed they could read it easily (see Supplementary Text for exam questions). Participants could start giving a verbal answer 10 s after question onset, indicated by the appearance of a countdown clock at the bottom of the screen (90 s per question; no time-outs were recorded). Data collected between questions and during the first 8 s of each question were truncated to avoid including non-question responses. Verbal responses to exam questions were anonymized and transcribed by two of the authors (M.M. and H.H.) using open source software (Audacity, www.audacityteam.org). Transcripts were then scored by two independent raters (teaching assistants on the course staff) on a scale of 0–3, and the mean was taken. Written exams taken by the students prior to scanning were rated in a similar way. The median score for experts was 85.4 out of 100 (range 69–100, s.d. 14.2). Expert responses rated below 2 were omitted from analyses (6 out of 64 responses in total). This was done to ensure expert brain activity patterns reflected correct answers. No student responses were omitted. Total exam scores (sum of all 16 questions) were normalized to a standard 0–100 scale.

**fMRI acquisition.** MRI data were collected on two 3-T full-body scanners (Siemens Skyra and Prisma) with 64 channel head coils. Scanner-participant pairing was kept constant throughout the experiment. Functional images were acquired using a T2*-weighted echo-planar imaging pulse sequence (TR 2000 ms, TE 28 ms, flip angle 80°, FOV 192 × 192 mm$^2$, whole-brain coverage with 38 transverse slices, 3 mm$^3$ voxels, no gap, GRAPPA iPAT 2). Anatomical images were acquired using a T1-weighted MPRAGE pulse sequence (1 mm$^3$ resolution).

**fMRI preprocessing.** Preprocessing was performed in FSL 6.0.1 (http://fsl.fmrib.ox.ac.uk/fsl), including slice time correction, motion correction, linear detrending, high-pass filtering (100 s cutoff), and gaussian smoothing (6 mm FWHM)[47,48]. Functional volumes were then coregistred and affine transformed to a template brain (MNI 152, Montreal Neurological Institute). Motion parameters (three translations and three rotations) were regressed out from functional data using linear regression. All calculations were performed in volume space. Data were analyzed using Python 3 (www.python.org) and R (www.r-project.org), using the Brain Imaging Analysis Kit (http://brainiak.org[49]) and custom code. Eight ROIs were anatomically defined using the probabilistic Harvard-Oxford cortical and subcortical structural atlases[50]. ROIs were defined across major DMN nodes in the angular gyrus, precuneus, and ACC, as well as in the hippocampus and posterior superior temporal gyrus, and control regions in early visual cortex (intracalcarine sulcus), early auditory cortex (Heschl's gyrus) and in subcortex (amygdala). Bilateral ROIs were created by taking the union of voxels in both hemispheres. A liberal threshold of >20% probability was used. To avoid circularity, all voxels within the anatomical mask were included and no functional data were used to define ROIs. Projections onto a cortical surface for visualization purposes were performed, as a final step, with Connectome Workbench[51].

**Alignment during lectures.** Multi-voxel BOLD patterns during lectures were obtained as follows. First, we used 30-s non-overlapping bins to extract multi-voxel activity (spatial alignment-to-class, Table 1). This yielded a single pattern for every bin in every participant. Then, to examine spatial similarities between participants during videos, we employed an inter-subject pattern correlation framework, which has been successfully used to uncover shared memory-related responses[21,22,28]. For each pattern in each student, we obtained an alignment-to-class measure by directly comparing the student pattern and the mean class pattern (average across all other students), using Pearson correlation. Then, correlation values were averaged within video segments. Throughout the manuscript, correlation values were transformed by Fisher's z prior to averaging and then back-transformed in order to minimize bias[52]. Finally, we averaged across segments to obtain a single alignment-to-class measure for every student during all lectures. Alignment was derived independently for each ROI and each searchlight. We then used alignment to predict student performance in the placement exam. To this end, we used a between-participants design, correlating alignment and overall exam scores (mean across questions). Statistical significance values were derived using a one-sided permutation test, with a null distribution created for each searchlight by shuffling score labels 1000 times.

Alternative spatial alignment-to-class measures (Supplementary Table 1) were calculated by changing the length of the time bin used. This was done to test the effects of our arbitrary choice of 30-s time bins for calculating alignment. We used 10-s non-overlapping bins (averaging over 5 TRs) and 2-s bins (calculating alignment in every TR). We also calculated a temporal alignment-to-class measure using inter-subject correlation (ISC). This was done by taking the mean of voxel time courses in each ROI in each student (mean across voxels, resulting in a single average time course) and correlating it with the mean class response (average across all other students, resulting in a single time course)[20,28].

**Power analysis across lectures.** We defined a "stable prediction index" across the cortex by considering the effect of information accumulation throughout the lectures on prediction success. To this end, we used the alignment-to-class values calculated for each one of our 21 individual lecture segments. We started by correlating exam scores with alignment-to-class in the first video (scan 1, segment 1). We then proceeded in sequence, correlating exam scores with the mean of alignment-to-class values across segments 1 and 2, and finally with the mean across all lecture segments. We performed this process for every cortical voxel using searchlight, to obtain a series of 21 r values for each voxel (one for each added segment). As before, a p value was calculated for each r value using a one-sided permutation test by randomizing score labels. Using a liberal threshold of p < 0.01 (uncorrected), we considered all voxels that showed a significant correlation between exam scores and alignment-to-class as calculated above. Lowering the threshold allowed us to include all potentially predictive voxels. For each voxel, we then defined the stable prediction index as the number of segments required to (i) reach a significant correlation between exam score and alignment and (ii) maintain significance for all subsequently added segments (no "breaks"). By design, a high index number (21) showed that data from all lecture segments were required to achieve a significant correlation and thus reflected late prediction. In contrast, a low index number (1) showed that significant prediction could be obtained by considering data from the first lecture segment alone, affording early prediction of exam score. Index values were calculated independently for each ROI and each searchlight.

**Correlation between alignment-to-class and alignment-to-experts.** Alignment-to-class during recaps was derived similarly to lectures. For each 30-s time bin in each student, we obtained an alignment-to-class measure by directly comparing the student pattern and the mean class pattern (average across all other students), using Pearson correlation. In addition, we obtained an alignment-to-experts measure by comparing the student pattern and the mean pattern across experts ("canonical" pattern). We thus obtained an alignment-to-class and alignment-to-

experts pattern for each student, in each bin. We used Pearson correlation to correlate these alignment measures using a between-participants design, obtaining a single correlation value in each time bin. Finally, we took the mean across all time bins within each recap, and then across recaps. Statistical significance values were derived using a one-sided permutation test, with a null distribution created for each time bin by shuffling student labels 1000 times. Correlation between alignment-to-class and alignment-to-experts during the exam was performed in an analogous manner, with questions used in place of time bins (see below).

**Neural alignment to experts during the exam**. Students' and experts' multi-voxel activity patterns during the exam were obtained by taking the mean fMRI BOLD signal during each question, in each participant. Each spatial pattern thus reflected neural responses associated with the specific subset of course topics included in that question. To compare student and class patterns, we again used the inter-subject pattern correlation framework. We derived an alignment-to-class score by correlating each question pattern in each student with the class average of the same question (mean across all other students) and then taking the mean across all students (Fig. 4a). We performed this on a question-by-question basis to obtain a vector of 16 alignment-to-class values for each student (one value for each question). Similarly, we derived an alignment-to-experts score by correlating each question pattern in each student with the mean pattern across experts, and obtained a vector of 16 alignment-to-expert values (one value for each question). Alignment was derived independently for each ROI and each searchlight. We then correlated alignment and question scores within students using Pearson correlation, obtaining a single *r*-value for each student. Finally, we took the mean across students. Statistical significance values were derived using a one-sided permutation test. We created a null distribution for each student by shuffling score labels 1000 times and then compared the mean across students to the mean null distribution.

**Knowledge structure alignment**. We defined "knowledge structures" as similarity matrices aimed at capturing the set of relationships between question representations. In the following, we describe how a student-specific knowledge structure was constructed and correlated with a class-derived template and an expert-derived template to derive (i) alignment-to-class and (ii) alignment-to-expert scores. Finally, we describe how within-participant correlation was used to examine the link between alignment and performance in individual students. First, we used the canonical class average and expert average patterns calculated for each question (see above) and constructed two templates. A class template was constructed by correlating "canonical class" question patterns with each other. This yielded a 16 question × 16 question symmetric similarity matrix comprising the distances between pairs of question patterns (*r* values) (Fig. 5a). Similarly, an expert template was constructed by correlating "canonical expert" question patterns with each other. This yielded two 16 question × 16 question symmetric similarity matrices comprising the distances between pairs of question patterns (*r* values). We then constructed a "knowledge structure" matrix for every student by correlating each question pattern (in that student) with the template pattern of all other questions. A single row in this structure thus represented the similarity between a student's neural response to a specific question and the template (class/expert) responses to every other question. Next, we correlated student and template matrices, row by row, excluding the diagonal, and obtained a question-by-question alignment score for each student. For each student, we thus derived a vector of 16 alignment-to-class values (one value for each question), and a vector of 16 alignment-to-expert values. Lastly, we correlated alignment and question scores within students using Pearson correlation, obtaining a single *r* value for each student, and took the mean across students. A null distribution was created for each student and a one-sided permutation test was used to determine statistical significance as before.

**Intersection analysis**. Intersection maps across data-driven searchlights were created by examining statistically significant voxels across analyses (*p* < 0.05, corrected). Figure 6a shows the intersection of the following maps: (i) correlation between alignment-to-class during lectures and exam scores (shown in Fig. 2d), (ii) correlation between alignment-to-class and alignment-to-experts during recaps (shown in Fig. 3c), (iii) correlation between same-question alignment-to-class during the final exam and exam score (shown in Fig. 4d, left panel), and (iv) correlation between knowledge structure alignment-to-class during the exam and exam score (shown in Fig. 5c). Figure 6b shows voxels in the intersection set of the following maps: (i) correlation between same-question alignment-to-class during the final exam and exam score (shown in Fig. 4d, left panel) and (ii) correlation between same-question alignment-to-experts during the final exam and exam score (shown in Fig. 4d, right panel).

**Reporting summary**. Further information on research design is available in the Nature Research Reporting Summary linked to this article.

## Data availability
The imaging data that support the findings of this study are available in OpenNeuro with the identifier https://doi.org/10.18112/openneuro.ds003233.v1.2.0 (ref. [53]). Source data are provided with this paper.

## Code availability
Analysis code is available on GitHub (https://github.com/me-sh/think_like_an_expert_paper).

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

## Acknowledgements

The authors wish to thank Princeton COS 126 staff and in particular Robert Sedgewick, Dan Leyzberg, Christopher Moretti, Kevin Wayne, Ibrahim Albluwi, Bridger Hahn, Thomas Schaffner, and Rachel Protacio; Mona Fixdal and the McGraw Center for Teaching and Learning; The Scully Center for the Neuroscience of Mind and Behavior; Peter J. Ramadge; and members of the Norman and Hasson labs for fruitful discussions. This study was supported by NIH Grant DP1-HD091948 to U.H. and by Intel Labs.

## Author contributions

M.M., L.H., H.H., K.A.N. and U.H. designed the experiment. M.M., L.H., H.H., Y.-F.L. and M.N. collected the data. M.M., K.A.N. and U.H. analyzed the data and wrote the manuscript.

## Competing interests

The authors declare no competing interests.
