## [Peer Review File · Nature Communications]

REVIEWER COMMENTS

Reviewer #1 (Remarks to the Author):

In this fMRI study, the authors characterize the neural representations of course-related content as college students progress throughout a computer science course, and relate the similarity of these representations to the central tendency of their classmates (and to a reference group of experts) to learning outcomes (e.g., exam performance). They also find that better-understood concepts evoked greater interpersonal neural alignment. This is an interesting paper that is sure to inspire interesting applications and future empirical work. Below, I outline some comments and suggestions that I think would be beneficial to consider in improving this manuscript.

- The paper mentions Cantlon and Li's very relevant work (Cantlon & Li, 2013, PLoS Biology) very briefly in the Discussion section. Given the exceptional relevance of this work, in terms of both its approach and findings (e.g., assessing neural response similarity between children learning math and adults who had learned math previously, while participants watched educational videos about math; using the similarity of children's neural responses to those of experts/adults to predict children's learning outcomes), it would greatly enrich the current paper to engage more deeply with Cantlon and Li's work – both with respect to their findings and approach - in the introduction and discussion sections. Doing so would better situate the study into the pertinent literature.

- In light of much of the authors' prior work that has characterized neural alignment in terms of similarity of time series of neural responses, it would be helpful for readers to be able to see how the current results change (or do not change) using such an approach, rather than the approach used in the current study's main analyses, which entail characterizing neural alignment in terms of similarities of spatial neural response patterns within particular time windows, then averaging across time. Reporting how results differ or converge using these different data analytic approaches would be helpful in informing subsequent research. It would also help to integrate the current study with much of the literature cited in the discussion establishing a link between shared understanding and neural alignment (often using time series correlations, rather than correlations of multivoxel response patterns, averaged across time).

- The sample size (N = 18 complete fMRI datasets; N = 20 when including participants with incomplete fMRI datasets) is very small, particularly for conducting correlational analyses between brain and behavior, for which significantly larger sample sizes have been widely recommended. While the design is longitudinal in that it involves scanning participants repeatedly over time, the analyses collapse data across sessions. Although this sample size has precedent in the authors' prior work, it is underpowered to detect all but very large effects. It would be helpful to at least discuss this limitation in the Discussion section.

- Relatedly, given the longitudinal nature of the fMRI study, it was surprising that changes over time were not examined. Did changes in understanding during the academic term relate to neural changes

(within participants) across scanning sessions?

- In the top panel of Fig. 3a and Fig. 3b, how was the example time bin chosen? It would be preferable to select a representative time bin, e.g., by choosing the time bin with the median correlation (amongst time bins) between alignment to experts and the class in each case. Otherwise, the results depicted at the top of Fig. 3a and Fig. 3b run the risk of appearing “cherry picked,” even though they may not be.

- Similarly, it would be preferable to replace the “example” participant visualized in the left panel of Fig. 5b (the red line) with data from the median participant (in terms of the participant’s correlation between question score and alignment to the class)

Minor concerns:

- How was the bin length of 30 s determined? Are the results robust to different parameter choices here?

- It would be helpful to proactively make code open to promote reproducibility, resolve any ambiguities in the description of methods, and promote the extension of this interesting line of inquiry by other researchers.

- It would be helpful to provide more explicit context and rationale for examining the relationships between neural alignment to the class and neural alignment to experts (p. 14). Presumably, the fact that these things are correlated has to do with the fact that the class was learning well. It would be interesting to discuss predictions about how interpretations would differ if alignment to classmates did not correlate with alignment to experts (e.g., in a class where the norm is to struggle with the material, rather than understand it well, or where particular erroneous thought patterns that would not characterize experts’ thinking would be common), in order to contextualize readers’ interpretations of these results

Reviewer #2 (Remarks to the Author):

In the manuscript “Think like an expert: Neural alignment predicts understanding in students taking an introduction to computer science course”, the authors conduct an ambitious multi-session fMRI scanning to examine learning in a real-life setting.

The manuscript is novel and the authors conduct an impressive amount of different types of analyses to examine neural signatures of individual and group level learning. The study will be of great interest to a wide range of readers of the journal. I feel that the manuscript is of high quality, yet it suffers from a few

issues that I hope the authors can address. If this is the case, I believe this manuscript will be then suitable for publication.

Major issues:

1. The authors need to avoid ambiguous terminology in the introduction. They should better define what type of “learning” they refer to and what they mean by “neural alignment”. In general, I am not sure that the term alignment is the correct noun to describe the phenomena studied by the authors – it is unclear to me what aligns/adjusts/calibrates here. Potentially correspondence is a better term, or something else that better describes the issue at hand.
2. The core findings of the study are contingent on the experts data. However, only minimal information is provided about the experts. Further details is needed on why graduate students are considered ‘experts’, how these ‘experts’ were selected, and additional demographic information about them (including their scores on the final exam). From the methods section, it seems as if there is some variance in the level of expertise of these ‘experts’.
3. The authors eloquently demonstrate a neural learning effect, centered around the DMN. However, an alternative explanation of the effect of ‘neural alignment’ on class performance can be just general adjustment to a college setting and getting familiar with how to perform in a standard class. What would have nailed the authors conclusions would have been a control condition, e.g., being asked throughout the multi-session scanning and final test questions on content not being learned. I realized such a control condition cannot be conducted at this timepoint. Nevertheless, I find the results novel and important. As such, the authors should address this issue in the limitation section in the discussion.
4. The authors compare a student to the class average – it is unclear whether in such a comparison that participant is excluded from the group average, similar to a leave one out cross validation. Since the authors are using Pearson’s correlation to examine the ‘neural alignment’, such an issue could inflate this correlation effect.
5. In relation to the knowledge structure analysis, the authors write that: “For each question in each participant, we measured the similarity of the neural pattern evoked by that question to the canonical patterns evoked by other questions (in the class average or in the experts). That is the knowledge structure for that question for that participant.” – it is unclear to me what the assumption of mental-neural alignment is based on.

Minor issues:

1. In the introduction, the authors should include additional relevant work on learning, research conducted at the neural and cognitive levels. Such work include the work of Marcello Matter, on the neural change in value learning, and that of Cynthia Siew, measuring conceptual change in learning using cognitive networks.
2. In the introduction, the authors write that: “For the most part, this body of work has examined well-established concept representations rather than newly acquired concepts.” – this should be expanded, as it is a core issue of the manuscript. In general, I think the introduction needs to be tightened. Furthermore, a general outline of the types of analyses conducted and described in the results section

would help in providing the reader the 'game plan'.

3. In comparing the neural patterns during questions – is question difficulty taken into consideration or somehow controlled?

4. The authors state that: "All students demonstrated knowledge gains (two-sided t-test, $t(19) = -12.6$, $p < 0.001$), with substantial variance across students (range 22-76 out of 100, median 53, s.d. 17.1)." – doesn't these low final scores show that students didn't learn that well at all?? Need to discuss this issue in the limitation section. Is this a consequence of being tested in the scanner?

5. The authors write that: "However, in the current work, alignment maps were not thresholded (i.e. statistical analysis for alignment effects was not performed), and all voxels were included in the subsequent searchlight analysis correlating alignment and exam scores" - the authors should explain why they are not thresholding their maps.

6. Table 2 – why report the amygdala if it has no significant relations across all comparisons?

7. Table 3:

a. Amygdala recap videos correlation is .09 – I fail to see how it leads to a significant p value.

b. Most regions have a much higher correlation with exam than with video recap – the authors need to discuss this. However, this is not true for AG or early visual regions – why? Finally, are these regions bilateral? If not need to indicate.

8. The authors write that: "These results indicated a key role for DMN regions across different phases of learning and further emphasized the link between alignment-to-experts and alignment-to-class measures." – this is a key finding of the manuscript – I think that it requires more unpacking and discussion.

9. In the beginning of the discussion, the authors write that: "The rapid changes in the field of education and the recent push towards online learning triggered by the recent pandemic have highlighted the need for novel teaching and assessment tools. In this work, we introduce a neural approach to predicting and assessing learning outcomes in real-life settings." However, given the complexity of the design, I fail to see how it can be easily applied to elucidate the effects of online learning. I would like the authors to further discuss this issue, which seems to be a key general issue resulting from this study.

10. In the discussion, the authors write that: "In the current study, we put forward the notion that understanding is mirrored in "neural alignment": the degree to which individual learners' neural representations match canonical representations observed in experts." This sentence perfectly defines the core issue of this study. It should be included in the introduction so that the readers understand what this study is all about.

11. Limitations paragraph is too short and should include additional issues raised here as well as those raised by the authors throughout the discussion.

12. Was the single non English speaker excluded? If not, why not?

13. Why were scans 2-5 videos accelerated?

Yoed Kenett

Reviewer #3 (Remarks to the Author):

The authors report a study that is very exciting, potentially very impactful to the fields of cognitive neuroscience and education, and which required a great deal of time and expertise to execute. It is a natural extension of prior work by several of the authors, and it is closely related to work that my lab has been conducting lately as well. Indeed, the central finding — that alignment between students and experts predicts real-life course-based learning across several topics — is exciting in its own right and it also holds promise for future endeavors of using neural data to measure learning in conjunction with traditional learning assessments. In particular, the use of neural data collected during the actual video lecture sessions to predict final exam performance is especially powerful and compelling.

The manuscript presents 3 complicated analysis procedures and describes them clearly despite their complexity. That said, as with any such analysis, there remain a few points that could be further clarified. I do not believe that these points in any way jeopardize the authors' central claims, and the authors have touched on these points to some degree already. However, I do believe that addressing them in more detail could provide important and useful information for the field. These points are summarized below. I would be happy to review a revised submission of the manuscript, if requested.

Signed,
David Kraemer

Several of my questions focus on interpreting and understanding what it means that alignment to a class of novice student peers is a consistently stronger predictor of learning than alignment to experts (despite the manuscript's title, "Think like an expert"). The discussion of this discrepancy could use some clarification:

1. Why is alignment-to-class consistently a better predictor than alignment-to-experts? The first set of results presented, which is also the study's strongest finding by far, is that neural pattern alignment among students during lectures is strongly predictive of final exam scores (e.g., $r = .75$ in the hippocampus). While this result is of great importance on its own, this analysis does not include experts at all. In fact, as shown in Table 3, for all ROIs the correlation between alignment-to-class values and alignment-to-expert values during the recap lectures is substantially lower on average than during the exam questions (e.g., in the ACC shown in Fig. 3, $r = .13$ for recap lectures and $r = .81$ for exam questions), although the authors do not address this discrepancy and instead cite both results as evidence that students think like experts (p. 30: "The tight link between alignment-to-class and alignment-to-experts suggests that students and experts may converge on a single set of shared neural states."). Similarly, for both analyses that use data collected during the exam period (the question analysis and the knowledge structures analysis), alignment-to-class is a stronger predictor of exam score than alignment-to-experts. In the case of the knowledge structure analysis, alignment to experts is not a significant predictor at all. While the authors describe this analysis as more noisy (which seems to be true), I think not enough discussion is given to the overall pattern that the strongest predictor of learning seems to be inter-subject correlation between novice learners during the lecture. Given the title of the manuscript ("Think like an expert") and the general framing of the paper (as well as the whole field of expertise research) as being focused on expertise and expert-like thinking, this

discrepancy bears further discussion and consideration. What do the authors make of the consistent pattern that alignment to novice peers is a better predictor of learning than alignment to experts?

2. Do the results reflect conceptual understanding or rote recall? The logic of each analyses makes sense, however the logic of each analysis taken in series is a bit harder to follow. In particular, I think the authors effectively motivate the idea of using alignment to the class average representation as a useful neural measure based on prior research and on the principle that the noise should “cancel out”. This approach seems almost like a central limit theorem of knowledge: amass enough independent observations and the sample mean should reflect the true representation of knowledge presented during the class. However, the logic of this approach is a bit in contrast to the logic of comparing the novice neural representations to the expert neural representations, which are supposed to represent the “canonical” (or true) information representations. In particular, this logic is most confusing when the results find (as I mentioned in point 1) that alignment to the class average is a *better* predictor than alignment to the experts. This seems to call for some consideration of an alternate explanation, e.g., maybe the students are paying more attention during lectures and remembering more lecture details (as has been shown with previous eeg studies mentioned in the discussion). In fact, the finding that the hippocampus and the “control regions” of primary visual and auditory cortex during the lectures are among the most predictive ROIs seems to support such a conclusion - that this alignment to class is demonstrating a classic subsequent memory effect, which is usually described in terms of rote mnemonic recall rather than conceptual understanding. The fact that the hippocampus shows the strongest effect is consistent with this interpretation as well. In contrast, rather than considering such a conclusion, the authors seem to assume that these patterns represent conceptual understanding per se. Given that alignment-to-experts during the exam seems to be a weaker predictor than alignment to other students during the lectures, do the authors think it is possible that their results reflect rote memorization (maybe due to increased attention) rather than conceptual understanding? Such a result would still be of great importance, and should at least be discussed either way.

3. Do the students’ neural data reflect common misconceptions? Given that the best performing student still missed nearly 1/4 of the questions on the final exam (p. 6 and Fig. 2), there is reason to believe that the average understanding of the class does not reflect true understanding of the concepts. The authors address this point by highlighting the (sometimes-observed) correlation between students neural patterns and experts neural patterns. Building on the 2 points mentioned above, the strongest evidence supporting the assertion that expert and student neural alignment predicts learning outcomes is the finding that during one of the analyses of exam questions (but not the other one), alignment-to-experts predicts exam scores (albeit not better than alignment-to-class). The authors interpret these findings on p. 18 as follows: “These findings show that neural alignment of specific question-by-question patterns was associated with better learning outcomes, indicating that concepts that were represented more similarly to the experts (and the class) were the concepts that students better understood.” I am inclined to agree with this interpretation, however, as noted above, focusing on alignment with experts doesn’t seem to tell the whole story (and the rest of the story is interesting!). In fact, despite the fact that much variance is not explained by the correlation between novices and experts (especially during lectures), the authors use these results to suggest that misconceptions shared among students could not

be contributing to the shared class representation. I'm not sure their data support such a conclusion. Again, consider the lecture periods, when the alignment-to-class neural data are most strongly predictive of exam performance. During the lectures, Table 2 reports that correlations between alignment-to-class and alignment-to-experts were $r = .28$ on average across all ROIs (or $.24$ without the "control ROIs"). Yet it was neural activity during the lectures (not the exams) that was most predictive of final exam performance (which again, was never even close to 100% perfect for any student). This pattern of results leaves open the possibility that the method presented here could be used to identify misconceptions, and times when the whole class is misunderstanding a concept presented in the lectures. This would be an incredibly useful finding, and in fact the study of misconceptions has been a major focus of STEM education research for decades. Are there any question items that were missed by the whole class (or most of the class) and for which many students gave a similar incorrect response? If so, then, foiling the authors' logic, neural alignment-to-class should also predict these incorrect answers. Again, such a finding could be a very important diagnostic tool for students and educators alike.

4. Do response times contribute to the correlation between the two neural measures (alignment-to-class and alignment-to-experts)? In terms of the data mentioned in point 3 (the exam period correlation between students and experts), notably, the authors address a critical alternative explanation for their findings by running a separate set of analyses (reported in the results and supplementary materials) to control for the effects of response time on exam question accuracy. As they put it on p. 19, this analysis "was motivated by the possibility that (i) longer answers might have yielded more stable spatial patterns, and that (ii) response length and quality could be linked (e.g. better answers could be longer)." Their solution was to regress response time out of the exam question scores to provide a measure of "residual scores" that are not correlated with response time. Importantly, they show that regressing out RT from the exam scores does not impact the results in terms of predicting exam scores with neural pattern correlations. This analysis effectively addresses the second point they raised (point ii) that "response length could be linked to response quality". However, because it is correcting the exam scores but not correcting the neural data, it does not seem to fully address the first point (point i) that "longer answers might have yielded more stable spatial patterns". In particular, could RT still have an effect on the correlation between alignment-to-class values and alignment-to-expert values (e.g., the results presented in Fig. 3)? This type of effect is sometimes accounted for by regressing RT from the neural data in the early stages of processing (although it's arguable that interesting neural variance would be lost by such a process). Especially during the self-paced exam questions it seems possible that the length of time spent on a response could affect the neural pattern (in stability or in other ways) and that this could contribute to the correlation between students and experts being stronger during the exam than during the lectures, as well as being more predictive of exam performance. If RT were found to contribute to these results, such a finding need not invalidate the results, however it could lead to a different interpretation of the alignment between students and experts, especially during the exam questions. What is the authors' take on this question?

Minor points:

- The "experts" were not experts in the traditional sense of the expertise literature (e.g., professors or

practicing professionals), but rather they were more advanced students than the novice learners. Given this point, further description of their mastery of the concepts would be useful. The methods section states that 6 items out of 64 were excluded because the experts' responses scored below 2. Given that alignment to the experts was a weaker correlate of exam performance (i.e., concept understanding) than other measures, it would be worth a comment about the possible differences between relying on these advanced students versus actual experts in the field. In terms of whether or not these graduate students displayed mastery of the target concepts, reporting a bit more detail about their scores (at least mean and sd of the experts' scores) would be useful.

- In the caption for Figure 4, there is a typo: "For all ROI analysis results, see table 3" - should read, "table 2"

- Clarification needed: In the two y-axis labels on Fig. 4C and on one of the plots in Fig. 5B the label reads "Question score (raw)" whereas the other similar plots read "Question score" and not "raw". Are these actually different or just inconsistently labeled? (I assume the latter.)

Again, despite the length of this review, I find the overall claims of the paper to be well supported and the main findings to be of great interest to the field. I would be happy to review a revised submission of the manuscript, if requested.

Signed,
David Kraemer

Reviewer #1 (Remarks to the Author):

- The paper mentions Cantlon and Li's very relevant work (Cantlon & Li, 2013, PLoS Biology) very briefly in the Discussion section. Given the exceptional relevance of this work, in terms of both its approach and findings (e.g., assessing neural response similarity between children learning math and adults who had learned math previously, while participants watched educational videos about math; using the similarity of children's neural responses to those of experts/adults to predict children's learning outcomes), it would greatly enrich the current paper to engage more deeply with Cantlon and Li's work – both with respect to their findings and approach - in the introduction and discussion sections. Doing so would better situate the study into the pertinent literature.

We thank the reviewer for highlighting the relevance of Cantlon and Li's paper to our work. As suggested, we now engage with it more deeply in the introduction as well as in the discussion. The amended text reads:

Introduction

“Furthermore, a study that compared neural activity time-courses in children and adults during short educational videos found that the degree to which children showed adult-like brain responses during math videos was correlated with their math test scores²³.”

Discussion

“Our finding that similarity to the class during lectures predicted performance is also in line with imaging results reported by Cantlon and Li²³. In that study, the authors used temporal ISC to show that children's scores in a standardized math test were correlated with the degree to which they showed adult-like brain responses during educational math videos. These correlations were localized to the intraparietal sulcus, a region previously implicated in numerical processing. We observed similar effects in our temporal ISC analysis as well as in our analysis of spatial patterns (Fig. 2, Fig. S1). The current study further extended the existing body of work to a real-world college course setting, enabling us to directly assess understanding of course-specific material from brain activity during learning and demonstrate a role for the DMN in learning, discussed below.”

- In light of much of the authors' prior work that has characterized neural alignment in terms of similarity of time series of neural responses, it would be helpful for readers to be able to see how the current results change (or do not change) using such an approach, rather than the approach used in the current study's main analyses, which entail characterizing neural alignment in terms of similarities of spatial neural response patterns within particular time windows, then averaging across time. Reporting how results differ or converge using these different data analytic approaches would be helpful in informing subsequent research. It would also help to integrate the current study with much of the literature cited in the discussion

establishing a link between shared understanding and neural alignment (often using time series correlations, rather than correlations of multivoxel response patterns, averaged across time).

We fully agree with the reviewer’s point. Note that temporal ISC analysis is possible only in cases when subjects process the same stimulus over time, e.g. during the lectures, when all subjects watched the exact same lectures. However, in cases where the events’ temporal structure differs across stimuli (as happens during the exam, where the length of answer can differ across subjects), we must use spatial ISC, which looks at the similarity of the average responses to each question. For completeness we now include the results of temporal inter-subject correlation (ISC) for the lectures portion of the study in our manuscript (Table S1). The amended text now reads:

Methods

“We also calculated a temporal alignment-to-class measure using Inter-Subject Correlation (ISC). This was done by taking the mean of voxel time courses in each ROI in each student (mean across voxels, resulting in a single average time course) and correlating it with the mean class response (average across all other students, resulting in a single time course).”

Results

“Alignment-to-class in ROIs during lecture videos showed a significant positive correlation with final exam scores in the angular gyrus, precuneus, anterior cingulate cortex (ACC) (all overlap with the DMN), and the hippocampus, as well as early visual and auditory areas (Fig. 2C-D, Table 2). Across ROIs, the highest correlation values were observed in the hippocampus, allowing the most reliable prediction of learning outcomes. Alternative measures of alignment-to-class, in which we varied the length of the time bin used for calculating spatial alignment, yielded similar though somewhat weaker results; likewise, we obtained similar results using a measure of shared responses in the time domain (temporal Inter-Subject Correlation, ISC; see Table S1 for results using these measures).”

Region of Interest (ROI)	Lectures		
	Corr. exam score and alignment-to-class (spatial) computed in 10s time bins	Corr. exam score and alignment-to-class (spatial) computed in each TR	Corr. exam score and alignment-to-class (temporal)
Angular Gyrus	0.51*	0.46*	0.59*
Ant. Cingulate (ACC)	0.50*	0.42 (n.s.)	0.36 (n.s.)
Hippocampus	0.59*	0.26 (n.s.)	0.61*
Post. Sup. Temporal Gyrus	0.30 (n.s.)	0.30 (n.s.)	0.51*
Precuneus	0.59*	0.59*	0.55*
Amygdala	0.35 (n.s.)	0.30 (n.s.)	0.49*
Early Auditory	0.54*	0.51*	0.39*
Early Visual	0.41 (n.s.)	0.42 (n.s.)	0.55*

Table S1. Prediction of exam score from alignment-to-class during lectures using alternative alignment-to-class measures. Left and middle columns, correlation of exam score and alignment-to-class computed using 10-second time bins (left) and with no binning (middle). Right column, correlation of exam score with a *temporal* measure of alignment-to-class (Inter-Subject Correlation, ISC). Results are shown in DMN ROIs as well as in control regions in sensory cortex (visual, intracalcarine cortex; auditory, Heschl's gyrus) and subcortex (amygdala). Green, significant correlation (permutation test, $p < 0.05$, FDR corrected across ROIs). * $p < 0.05$, ** $p < 0.01$, n.s., not significant.

- The sample size (N = 18 complete fMRI datasets; N = 20 when including participants with incomplete fMRI datasets) is very small, particularly for conducting correlational analyses between brain and behavior, for which significantly larger sample sizes have been widely recommended. While the design is longitudinal in that it involves scanning participants repeatedly over time, the analyses collapse data across sessions. Although this sample size has precedent in the authors' prior work, it is underpowered to detect all but very large effects. It would be helpful to at least discuss this limitation in the Discussion section.

The reviewer's point about sample size is well taken. In this study, we collected an unusually large amount of data (> 3 hours of functional scans, nearly 6000 TRs) per participant, which limited the number of participants that we could run. The very large amount of data per participant puts us in a favorable position (in terms of signal-to-noise) relative to other studies that used a similar number of participants but collected less data per participant. However, the limited number of participants may have reduced our power to detect smaller effects, as the reviewer suggests. To assess how the effects reported here depend on the amount of data that was collected, we ran a power analysis (figure S1) in which we simulated a situation where scanning was halted after a single lecture segment, two segments and so forth. This analysis showed that, in some regions, correlations between neural alignment (during lectures) and exam performance were robust even with a limited amount of fMRI data, but effects in other regions only became significant with much larger amounts of data, and it is possible that even more regions would have showed significant effects if we had tested more participants or collected more data per participant. As suggested, we now address the sample size in the limitation part of the discussion. The amended text reads:

Limitations

"Another limitation concerns the relatively small number of participants: our longitudinal design allowed us to collect a large amount of data (> 3 hours of functional scans) per student participant, but limited the number of participants that we could run given available scanning resources. Although the number of student participants in the current study is in line with previous studies from our group^{21,43,44}, it may have limited our ability to detect smaller effects. To determine how much lecture data was required to obtain robust correlations with exam scores, we conducted a power analysis (Supplementary Results and Fig. S1) that simulated a situation where scanning was halted after a single lecture segment, two segments and so forth.

Importantly, while some cortical regions in the DMN showed a robust correlation between alignment-to-class and exam scores given relatively small amounts of data, other cortical regions only showed a robust correlation given the entire dataset. It is possible that even more regions would have shown significant effects if we had tested more participants or collected more data per participant. The silver lining of this result is that simpler and shorter experimental designs than those employed here may be sufficient to study real-world learning effects in DMN regions.”

- Relatedly, given the longitudinal nature of the fMRI study, it was surprising that changes over time were not examined. Did changes in understanding during the academic term relate to neural changes (within participants) across scanning sessions?

We thank the reviewer for raising this issue. In the current study, we used a measure of understanding taken at a single time point (final exam) rather than at multiple time points during the academic term. The dataset analyzed here therefore does not allow examining changes in understanding during the academic term as such. However, the reviewer raises a promising direction for future work. We now address this issue in the text.

Discussion

“A promising direction for future work lies in leveraging recent methodological advances (e.g., in language modeling and network science methods) to track the development of students’ knowledge structure during learning.”

- In the top panel of Fig. 3a and Fig. 3b, how was the example time bin chosen? It would be preferable to select a representative time bin, e.g., by choosing the time bin with the median correlation (amongst time bins) between alignment to experts and the class in each case. Otherwise, the results depicted at the top of Fig. 3a and Fig. 3b run the risk of appearing “cherry picked,” even though they may not be.

- Similarly, it would be preferable to replace the “example” participant visualized in the left panel of Fig. 5b (the red line) with data from the median participant (in terms of the participant’s correlation between question score and alignment to the class)

We thank the reviewer for this opportunity to discuss our point of view. Examples were chosen for maximum clarity with the aim of conveying how the analysis was performed. These examples are presented right next to summary panels where responses from all participants are shown (grey lines) along with a summary statistic (mean correlation), in order to convey all pertinent information and avoid any appearance of “cherry picking”. We further wished to enhance transparency by maintaining consistency across panels and figures. To this end, in Fig. 4 and Fig. 5 we show data from the same participant (“participant 1”). As the reviewer pointed out, this participant showed the highest correlation in panel 5b. However, this is not the case in figure 4 for both alignment-to-class and alignment-to-experts. We worry that selecting a

different participant for each panel might be less consistent and make the examples less clear. However, we made sure to highlight the fact that these are in fact examples.

Minor concerns:

- How was the bin length of 30 s determined? Are the results robust to different parameter choices here?

We thank the reviewer for this question. Alignment was somewhat robust to changes in bin size, with significant effects found for 10-second bins (averaging over 5 TRs) across most ROIs. Effects for 2-second bins (calculating alignment in every TR) were noticeably weaker. We also added appropriate references to the results and methods. We now include results for different parameter choices in the manuscript (Table S1).

Results

“Alternative measures of alignment-to-class, in which we varied the length of the time bin used for calculating spatial alignment, yielded similar though somewhat weaker results”. (Table S1)

Methods

“Alternative spatial alignment-to-class measures (Table S1) were calculated by changing the length of the time bin used. This was done to test the effects of our arbitrary choice of 30-second time bins for calculating alignment. We used 10-second non-overlapping bins (averaging over 5 TRs) and 2-second bins (calculating alignment in every TR).”

Region of Interest (ROI)	Lectures		
	Corr. exam score and alignment-to-class (spatial) computed in 10s time bins	Corr. exam score and alignment-to-class (spatial) computed in each TR	Corr. exam score and alignment-to-class (temporal)
Angular Gyrus	0.51*	0.46*	0.59*
Ant. Cingulate (ACC)	0.50*	0.42 (n.s.)	0.36 (n.s.)
Hippocampus	0.59*	0.26 (n.s.)	0.61*
Post. Sup. Temporal Gyrus	0.30 (n.s.)	0.30 (n.s.)	0.51*
Precuneus	0.59*	0.59*	0.55*
Amygdala	0.35 (n.s.)	0.30 (n.s.)	0.49*
Early Auditory	0.54*	0.51*	0.39*
Early Visual	0.41 (n.s.)	0.42 (n.s.)	0.55*

Table S1. Prediction of exam score from alignment-to-class during lectures using alternative alignment-to-class measures. Left and middle columns, correlation of exam score and alignment-to-class computed using 10-second time bins (left) and with no binning (middle). Right column, correlation of exam score with a *temporal* measure of alignment-to-class (Inter-Subject Correlation, ISC). Results are shown in DMN ROIs as well as in control regions in sensory cortex (visual, intracalcarine cortex; auditory, Heschl's gyrus) and subcortex

(amygdala). Green, significant correlation (permutation test, $p < 0.05$, FDR corrected across ROIs). * $p < 0.05$, ** $p < 0.01$, n.s., not significant.

- It would be helpful to proactively make code open to promote reproducibility, resolve any ambiguities in the description of methods, and promote the extension of this interesting line of inquiry by other researchers.

We thank the reviewer for this important comment, and agree for the need to provide the reader with access to the data and code to obtain full transparency. As suggested, we have made the analysis code publicly available through GitHub and amended the resources availability statement accordingly.

- It would be helpful to provide more explicit context and rationale for examining the relationships between neural alignment to the class and neural alignment to experts (p. 14). Presumably, the fact that these things are correlated has to do with the fact that the class was learning well. It would be interesting to discuss predictions about how interpretations would differ if alignment to classmates did not correlate with alignment to experts (e.g., in a class where the norm is to struggle with the material, rather than understand it well, or where particular erroneous thought patterns that would not characterize experts' thinking would be common), in order to contextualize readers' interpretations of these results

We thank the reviewer for these insightful suggestions. We amended the text to provide more explicit context and rationale for examining the relationships between neural alignment to the class and neural alignment to experts. The amended text reads:

Results

“Neural alignment-to-class was strongly correlated with alignment-to-experts. Experts were scanned during recap videos (16 minutes in total) and while taking the final exam. We separately calculated alignment-to-class and alignment-to-experts for each student in each task and then correlated these measures using a between-participants design (see Methods). The goal of this analysis was to examine whether alignment-to-class reflected convergence on expert patterns.”

Discussion (under “Class patterns reflect expert patterns”)

“According to this view, when individual patterns are averaged and idiosyncratic differences cancel out, what emerges is a good approximation of an ideal “canonical” representation. We would like to suggest this is analogous to a “central limit theorem” of knowledge: the mean is a reflection of the fact that most students, most of the time, follow the lecture as intended: what they share is the correct interpretation of course material. On the other hand, in a class where the norm is to struggle with the material, rather than understand it well, common response patterns may not emerge. Another caveat is that common misunderstandings would also be reflected in the common pattern. These misunderstandings, however, would not be shared by experts, resulting in high alignment across students but low alignment between students and experts.”

Reviewer #2 (Remarks to the Author):

In the manuscript “Think like an expert: Neural alignment predicts understanding in students taking an introduction to computer science course”, the authors conduct an ambitious multi-session fMRI scanning to examine learning in a real-life setting.

The manuscript is novel and the authors conduct an impressive amount of different types of analyses to examine neural signatures of individual and group level learning. The study will be of great interest to a wide range of readers of the journal. I feel that the manuscript is of high quality, yet it suffers from a few issues that I hope the authors can address. If this is the case, I believe this manuscript will be then suitable for publication.

Major issues:

1. The authors need to avoid ambiguous terminology in the introduction. They should better define what type of “learning” they refer to and what they mean by “neural alignment”. In general, I am not sure that the term alignment is the correct noun to describe the phenomena studied by the authors – it is unclear to me what aligns/adjusts/calibrates here. Potentially correspondence is a better term, or something else that better describes the issue at hand.

We thank the reviewer for pointing out this ambiguity in terms. We amended the text with the aim of clarifying that we refer to STEM learning in academia and stated our hypothesis more clearly by providing a definition of neural alignment as suggested. The amended text reads:

Introduction

“Here, we focused on STEM learning in academia. Our goal was to use shared neural activity patterns across learners and experts to quantify and predict understanding in a popular course at Princeton University. We tested the hypothesis that understanding is mirrored in “neural alignment”: the degree to which individual learners’ neural representations match canonical representations observed in experts.”

2. The core findings of the study are contingent on the experts data. However, only minimal information is provided about the experts. Further details is needed on why graduate students are considered ‘experts’, how these ‘experts’ were selected, and additional demographic information about them (including their scores on the final exam). From the methods section, it seems as if there is some variance in the level of expertise of these ‘experts’.

As suggested, we now include more information about the experts and their performance:

- a. Under participants and stimuli: “Experts all had an undergraduate or graduate degree in computer science and reported having significant programming experience as well as knowledge of introduction to computer science course material (≥ 6 on a 7-point Likert scale).”
- b. Under experimental procedures: “The median score for experts was 85 out of 100 (range 69-100, s.d. 14.2).”

3. The authors eloquently demonstrate a neural learning effect, centered around the DMN. However, an alternative explanation of the effect of ‘neural alignment’ on class performance can be just general adjustment to a college setting and getting familiar with how to perform in a standard class. What would have nailed the authors conclusions would have been a control condition, e.g., being asked throughout the multi-session scanning and final test questions on content not being learned. I realized such a control condition cannot be conducted at this timepoint. Nevertheless, I find the results novel and important. As such, the authors should address this issue in the limitation section in the discussion.

- As the reviewer correctly points out, neural alignment during the lectures may reflect getting familiar with how to perform in a standard class as well as other factors in addition to those the manuscript focuses on. As suggested, we now address this in the limitations section (see below).

Limitations

“Finally, like other neural measures, alignment between individuals likely reflects multiple contributing factors. These could include similarities in students’ educational background and familiarity with the teaching method. Mapping the different factors and their unique contributions remains a topic for future work.”

- We also wish to point out that the analyses of neural responses during the end-of-course exam (relating same-question patterns and knowledge structure patterns to performance on specific exam questions; Fig. 4 and Fig. 5) were conducted within-subjects, and thus can not be explained in terms of individual differences. We address this in more detail now in the discussion.

Discussion

“Together with our same-question results, the positive correlation between knowledge-structure alignment and exam performance provides key support for our alignment-as-understanding hypothesis. The correlation between alignment-to-class during lectures and exam performance, discussed above, could be explained in terms of coarse-grained individual-difference variables, even if there is no direct link between alignment-to-class and understanding (e.g., conscientious

students may show high alignment-to-class during lectures because they closely attend to videos, and obtain good exam scores because they study a lot outside of class, leading to a correlation between alignment-to-class and exam performance). Crucially, the analyses relating same-question alignment and knowledge-structure alignment to performance on specific exam questions were conducted within-subjects, and thus the results of these analyses cannot be explained in terms of differences between participants.”

4. The authors compare a student to the class average – it is unclear whether in such a comparison that participant is excluded from the group average, similar to a leave one out cross validation. Since the authors are using Pearson’s correlation to examine the ‘neural alignment’, such an issue could inflate this correlation effect.

We thank the reviewer for allowing us to clarify this critical point. Alignment-to-class is calculated by comparing each student’s response patterns to the mean response patterns across **all other** students. This ensures that correlation effects are not inflated. We have now highlighted this throughout the manuscript as well as in the newly added parts dealing with temporal Inter-Subject Correlation (ISC), which is calculated in a similar way.

5. In relation to the knowledge structure analysis, the authors write that: “For each question in each participant, we measured the similarity of the neural pattern evoked by that question to the canonical patterns evoked by other questions (in the class average or in the experts). That is the knowledge structure for that question for that participant.” – it is unclear to me what the assumption of mental-neural alignment is based on.

We regret the confusing language here. We wish to make no assumption at this point in the text, merely to describe how we defined the neural knowledge structure. The relationship between this structure and behavior is explored in detail later on. Accordingly, we amended the text to read as follows:

Results

“For each question in each participant, we measured the similarity of the neural pattern evoked by that question to the canonical patterns evoked by *other* questions (in the class average or in the experts). We defined the set of question-specific relationships as the “knowledge structure” for that question for that participant.”

Minor issues:

1. In the introduction, the authors should include additional relevant work on learning, research conducted at the neural and cognitive levels. Such work include the work of Marcello Matter, on the neural change in value learning, and that of Cynthia Siew, measuring conceptual change in learning using cognitive networks.

We thank the reviewer for pointing us towards this relevant literature. We now cite it in the Discussion:

“A promising direction for future work lies in leveraging recent methodological advances (e.g., in language modeling and network science methods) to track the development of students’ knowledge structure during learning^{36–38}. These methods could make it possible to delineate the exact relationship between student and expert knowledge structures in the DMN”.

References

36. Siew, C. S. Q. Applications of Network Science to Education Research: Quantifying Knowledge and the Development of Expertise through Network Analysis. *Education Sciences* 10, 101 (2020).
37. Bassett, D. S. & Mattar, M. G. A Network Neuroscience of Human Learning: Potential to Inform Quantitative Theories of Brain and Behavior. *Trends in Cognitive Sciences* 21, 250–264 (2017).
38. Kenett, Y. N., Betzel, R. F. & Beaty, R. E. Community structure of the creative brain at rest. *NeuroImage* 210, 116578 (2020).

2. In the introduction, the authors write that: “For the most part, this body of work has examined well-established concept representations rather than newly acquired concepts.” – this should be expanded, as it is a core issue of the manuscript.

In general, I think the introduction needs to be tightened. Furthermore, a general outline of the types of analyses conducted and described in the results section would help in providing the reader the ‘game plan’.

We thank the reviewer for these helpful comments. We now give concrete examples and literature references for prior work on well-established concept representations. The amended text reads: “For the most part, this body of work has examined well-established concept representations (e.g. representations of objects and animals^{10–12}) rather than newly acquired concepts”.

References

10. Haxby, J. V. et al. Distributed and overlapping representations of faces and objects in ventral temporal cortex. *Science* 293, 2425–30 (2001).
11. Connolly, A. C. et al. The Representation of Biological Classes in the Human Brain. *J. Neurosci.* 32, 2608–2618 (2012).
12. Kriegeskorte, N., Mur, M. & Bandettini, P. A. Representational similarity analysis - connecting the branches of systems neuroscience. *Front. Syst. Neurosci.* 2, (2008).

In addition, the final paragraph of the introduction now includes specific references to the figures in order to provide the reader with an outline of the manuscript, as suggested.

3. In comparing the neural patterns during questions – is question difficulty taken into consideration or somehow controlled?

We fully agree that question difficulty could affect neural patterns. Unfortunately, it was not feasible to get a per-question measure of difficulty for this study; this would require administering the exam to a large, separate cohort of students at the end of the semester, but -- as noted in our paper -- the exam we used in our study was a placement exam routinely administered at the *start* of the course (and not the end), and scores from the start of the course would not accurately reflect question difficulty. Having said this, it is possible to gain some insight into this issue based on our treatment of response length, which often shows a positive relationship with question difficulty. As described in the paper, our key results using the exam-period fMRI data still hold up when we control for response length.

Following this and previous comments, we now directly recognize the possible contributions of other factors on our results in the limitations section: “Finally, like other neural measures, alignment between individuals likely reflects multiple contributing factors. These could include similarities in students’ educational background and familiarity with the teaching method. Mapping the different factors and their unique contributions remains a topic for future work”.

4. The authors state that: “All students demonstrated knowledge gains (two-sided t-test, $t(19) = -12.6$, $p < 0.001$), with substantial variance across students (range 22-76 out of 100, median 53, s.d. 17.1).” – doesn’t these low final scores show that students didn’t learn that well at all?? Need to discuss this issue in the limitation section. Is this a consequence of being tested in the scanner?

We regret the ambiguity in the original text. The reported scores represent pre-post changes, as the baseline/“pre” exam was the *same exam* as the final/“post” exam (the same questions in the same order). That is, the two columns shown in Fig. 1b reflect scores in the same exam (all students received a score of zero on the baseline exam). Therefore, while the scores in the final exam were relatively low, they represent significant knowledge gains. We also wish to point out that the difficulty level of the exam was out of our hands as we used the actual placement exam compiled by course staff. Nevertheless, we got a wide range and high variance of student scores, which is desirable from our point of view (e.g. avoiding ceiling and floor effects). To resolve the ambiguity, we amended the manuscript to read:

Results

“To establish a baseline, the *same exam* was also given to students at the beginning of the semester, in written form (“pre” exam)...By the end of the course, all students demonstrated knowledge gains (pre-post comparison, two-sided t-test, $t(19) = -12.6$, $p < 0.001$).”

5. The authors write that: “However, in the current work, alignment maps were not thresholded (i.e. statistical analysis for alignment effects was not performed), and all voxels were included in

the subsequent searchlight analysis correlating alignment and exam scores” - the authors should explain why they are not thresholding their maps.

This is a crucial point and we thank the reviewer for the opportunity to clarify it. All effect maps included in the manuscript were properly thresholded using a permutation test while controlling for the false discovery rate. The reviewer is referring to the map shown in fig 2B, for which no statistical test was performed. We included it in order to demonstrate the analysis flow where we first calculate alignment-to-class values for every voxel in the cortex, and then correlate alignment-to-class and exam scores. The quoted sentence from the manuscript is simply stating that we included all voxels - with both high and low ISC values - in order to correlate this variance with the variance in behavior. This would not be possible had we used a voxel-wise threshold as a pre-selection step (e.g. as in Baldassano et al., Event Structure in Continuous Narrative Perception and Memory. *Neuron*. 2017.)

To resolve any ambiguity, we now amended the text to read:

Results

“The alignment map was qualitatively in line with the body of literature showing that watching the same video elicits shared activity patterns across individuals^{20,28}. However, in the current work, alignment maps were not thresholded (i.e. statistical analysis for alignment effects was not performed), and all voxels were included in the subsequent searchlight analysis in which we correlated alignment and exam scores. This was done in order to test whether variance in alignment was related to variance in scores and avoid excluding brain regions (e.g. the hippocampus) where alignment values were lower yet could be predictive of learning outcomes.

6. Table 2 – why report the amygdala if it has no significant relations across all comparisons?

The amygdala ROI was pre-selected as a sub-cortical control region, where we did not expect to find significant prediction effects. We also selected two control regions in sensory cortex (visual, intracalcarine cortex; auditory, Heschl's gyrus). Unexpectedly, in the cortical control regions, we found significant prediction effects for the lectures, which we discuss in the last paragraph of the Discussion.

To clarify this, we amended the text in the Results to read:

“Correlation between alignment and exam scores was done using a between-participants design, first in eight anatomically-defined regions of interest (ROIs) and then across the entire cerebral cortex using a searchlight analysis. Our ROIs included major nodes of the DMN and the hippocampus as well as control regions in early sensory cortex and the amygdala. Our selection of ROIs was motivated by findings that activity in the DMN during memory encoding of new content (real-life stories or audio visual movies) predicted recall success for that material^{21,22,29}”

7. Table 3:

- a. Amygdala recap videos correlation is .09 – I fail to see how it leads to a significant p value.
- b. Most regions have a much higher correlation with exam than with video recap – the authors need to discuss this. However, this is not true for AG or early visual regions – why? Finally, are these regions bilateral? If not need to indicate.

a. We thank the reviewer for calling attention to the small effect size in the Amygdala. We re-ran the analysis in order to verify the result and were able to confirm that this value is significantly above 0 after correcting for multiple comparisons across ROIs. Detecting such small effects was made possible in this case thanks to the high sensitivity of the analysis, i.e. correlating alignment-to-class and alignment-to-experts values in *each time bin* across students. Our permutation test called for creating a distribution of null values through shuffling student labels. These null values were very close to 0, enabling us to detect small positive effects such as those reported here. We include an in-depth description of this procedure in the Methods section.

b. We thank the reviewer for pointing out the higher correlation with exam than recap videos in some ROIs. We agree that these differences in effect size and the evident variance between cortical regions are potentially of great interest. We speculate that they could be related to the very different nature of these two conditions (passive viewing of videos vs. active generation of verbal responses). The amended text is below.

Discussion

“The tight link between alignment-to-class and alignment-to-experts suggests that students and experts may converge on a single set of shared neural states. However, we observed substantial variability in correlation magnitude between these two alignment measures across cortical regions and between tasks (Table 3). We speculate that these differences could be linked to the different nature of the tasks (passive viewing of recap videos and active generation of verbal responses during the exam). Nevertheless, the factors contributing to this variability remain unknown at this point.”

- c. All ROIs were defined bilaterally (Methods, under “fMRI processing”).

8. The authors write that: “These results indicated a key role for DMN regions across different phases of learning and further emphasized the link between alignment-to-experts and alignment-to-class measures.” – this is a key finding of the manuscript – I think that it requires more unpacking and discussion.

We thank the reviewer for pointing out that this statement requires unpacking. We now expand on this statement in the discussion under “A key role for medial DMN regions during learning”. The amended text reads:

Discussion

“The significance of DMN cortical structures in our results highlights the role these regions play during key stages of the learning process, from first exposure to course video lectures to review of learned material (recap videos) and, finally, question answering during the exam (Fig. 6). Effects in these regions were robust, emerging for both alignment measures (to class and to experts). These findings are in line with previous work that localized behaviorally-relevant, memory-related shared representations to these areas^{21,22}. They are also consistent with earlier findings that specific patterns of activity during memory encoding in DMN regions predicted recall performance²⁹, as well as with a recent report of hippocampal changes triggered by learning the structures and names of organic compounds.”

9. In the beginning of the discussion, the authors write that: “The rapid changes in the field of education and the recent push towards online learning triggered by the recent pandemic have highlighted the need for novel teaching and assessment tools. In this work, we introduce a neural approach to predicting and assessing learning outcomes in real-life settings.” However, given the complexity of the design, I fail to see how it can be easily applied to elucidate the effects of online learning. I would like the authors to further discuss this issue, which seems to be a key general issue resulting from this study.

The reviewer is right to point out that our multisession fMRI design is too cumbersome to use to do practical, real-world assessment of learning. Rather, the goal of our basic science fMRI study was to discover underlying principles of how to assess understanding based on neural data. Such findings perhaps could help in the future in building more portable tools to assess learning in class. As a concrete example, in our power analysis (Fig. S1), we assessed how much data was needed to obtain robust prediction effects (relating neural alignment during lectures to exam scores) in different brain areas. This analysis pointed to several regions where significant prediction could be obtained with limited data. Future, assessment-oriented studies could focus on these regions using simpler (and shorter) designs. We now address this issue directly in the Discussion:

Limitations

“...we conducted a power analysis (Supplementary Results and Fig. S1) that simulated a situation where scanning was halted after a single lecture segment, two segments and so forth. Importantly, while some cortical regions in the DMN showed a robust correlation between alignment-to-class and exam scores given relatively small amounts of data, other cortical regions only showed a robust correlation given the entire dataset. It is possible that even more regions would have shown significant effects if we had tested more participants or collected more data per participant. The silver lining of this result is that simpler and shorter experimental designs than those employed here may be sufficient to study real-world learning effects in DMN regions”.

10. In the discussion, the authors write that: “In the current study, we put forward the notion that understanding is mirrored in “neural alignment”: the degree to which individual learners’ neural representations match canonical representations observed in experts.” This sentence perfectly

defines the core issue of this study. It should be included in the introduction so that the readers understand what this study is all about.

We amended the Introduction section to include this statement.

11. Limitations paragraph is too short and should include additional issues raised here as well as those raised by the authors throughout the discussion.

The revised manuscript now includes a more comprehensive limitations section.

12. Was the single non English speaker excluded? If not, why not?

We took an extremely conservative approach in including the non-English native participant in our analyses. As an expert, his neural responses were used in aggregate with other experts to create the experts' template. Any discrepancy between his neural responses and the other experts' responses could only serve to make that template less uniform (more "noisy"), interfering with our analyses. Therefore the more conservative choice in this case was to include this data in our analyses.

13. Why were scans 2-5 videos accelerated?

We now include this information in the text:

Experimental Procedures

"Participants were asked to watch lecture videos as they normally would. Lecture videos were shown at normal speed (first scan) or slightly accelerated (x1.15 speed, scans 2-5, accelerated for all participants at their request)."

Yoed Kenett

Reviewer #3 (Remarks to the Author):

The authors report a study that is very exciting, potentially very impactful to the fields of cognitive neuroscience and education, and which required a great deal of time and expertise to execute. It is a natural extension of prior work by several of the authors, and it is closely related to work that my lab has been conducting lately as well. Indeed, the central finding — that alignment between students and experts predicts real-life course-based learning across several topics — is exciting in its own right and it also holds promise for future endeavors of using neural data to measure learning in conjunction with traditional learning assessments. In particular, the use of neural data collected during the actual video lecture sessions to predict final exam performance is especially powerful and compelling.

The manuscript presents 3 complicated analysis procedures and describes them clearly despite their complexity. That said, as with any such analysis, there remain a few points that could be further clarified. I do not believe that these points in any way jeopardize the authors' central claims, and the authors have touched on these points to some degree already. However, I do believe that addressing them in more detail could provide important and useful information for the field. These points are summarized below. I would be happy to review a revised submission of the manuscript, if requested.

Signed,

David Kraemer

Several of my questions focus on interpreting and understanding what it means that alignment to a class of novice student peers is a consistently stronger predictor of learning than alignment to experts (despite the manuscript's title, "Think like an expert"). The discussion of this discrepancy could use some clarification:

1. Why is alignment-to-class consistently a better predictor than alignment-to-experts? The first set of results presented, which is also the study's strongest finding by far, is that neural pattern alignment among students during lectures is strongly predictive of final exam scores (e.g., $r = .75$ in the hippocampus). While this result is of great importance on its own, this analysis does not include experts at all. In fact, as shown in Table 3, for all ROIs the correlation between alignment-to-class values and alignment-to-expert values during the recap lectures is substantially lower on average than during the exam questions (e.g., in the ACC shown in Fig. 3, $r = .13$ for recap lectures and $r = .81$ for exam questions), although the authors do not address this discrepancy and instead cite both results as evidence that students think like experts (p. 30: "The tight link between alignment-to-class and alignment-to-experts suggests that students and experts may converge on a single set of shared neural states."). Similarly, for both analyses that use data collected during the exam period (the question analysis and the knowledge structures analysis), alignment-to-class is a stronger predictor of exam score than alignment-to-experts. In the case of the knowledge structure analysis, alignment to experts is not a significant predictor at all. While the authors describe this analysis as more noisy (which seems to be true), I think not enough discussion is given to the overall pattern that the strongest predictor of learning seems to be inter-subject correlation between novice learners during the lecture. Given the title of the manuscript ("Think like an expert") and the general framing of the paper (as well as the whole field of expertise research) as being focused on expertise and expert-like thinking, this discrepancy bears further discussion and consideration. What do the authors make of the consistent pattern that alignment to novice peers is a better predictor of learning than alignment to experts?

This is a fundamental point in the manuscript and we thank the reviewer for highlighting it. The complex relationship we found between alignment-to-class and alignment-to-experts is indeed central to this work. First, we acknowledge that the reviewer is right in pointing that alignment-

to-class is a better predictor for learning outcome than alignment-to-experts in our study. There are two possible explanations for this difference. One possible explanation relates to the fact that we could only scan five experts, which drastically reduced the power of the student-to-experts analyses relative to the student-to-other-students analyses. The fact that we could still replicate the effect seen within students with only five experts may actually point to the robustness of it, and may suggest that the student-experts effect is stronger than may seem by looking at absolute correlation values. The other possible explanation of the difference between the alignment-to-class and alignment-to-experts results is that there is a substantive difference in the patterns from the two groups (e.g., experts' grasp of course material may be more holistic, drawing on their broader understanding of the field, and therefore qualitatively different from that of students). To acknowledge these issues, we added a section to the Discussion that addresses possible interpretations for these differences in predictive power between alignment-to-class and alignment-to-experts:

Discussion (under "Learning the right 'knowledge structure'"):

"Our results further show that, even at the end of the course, students' degree of knowledge structure convergence to experts did not predict students' question-answering accuracy. A likely cause of this null result is lack of power (fewer experts than students in our dataset). However, an alternative explanation is that alignment to novice peers is actually a better measure of understanding than alignment to experts, even when power is matched between these measures. This could also account for the stronger student-to-class effects found in our same-question analysis (Table 2, Fig. 4D). One reason for this could be that experts' grasp of course material is more holistic, drawing on their broader understanding of the field, and therefore qualitatively different from that of students. In this case, the class average could provide a unique window into the neural state best associated with successful learning. Resolving this issue would require conducting power-matched comparisons between students and experts. However, if this hypothesis is substantiated, it would call for a reconsideration of the traditional focus on expert-like thinking in expertise research in favor of group-like thinking among learners."

We also amended the following text:

Discussion (under "Class patterns reflect expert patterns"):

"However, we observed substantial variability in correlation magnitude between these two alignment measures across cortical regions and between tasks (Table 3). We speculate that these differences could be linked to the different nature of the tasks (passive viewing of recap videos and active generation of verbal responses during the exam). Nevertheless, the factors contributing to this variability remain unknown at this point."

2. Do the results reflect conceptual understanding or rote recall? The logic of each analyses makes sense, however the logic of each analysis taken in series is a bit harder to follow. In particular, I think the authors effectively motivate the idea of using alignment to the class average representation as a useful neural measure based on prior research and on the principle

that the noise should “cancel out”. This approach seems almost like a central limit theorem of knowledge: amass enough independent observations and the sample mean should reflect the true representation of knowledge presented during the class. However, the logic of this approach is a bit in contrast to the logic of comparing the novice neural representations to the expert neural representations, which are supposed to represent the “canonical” (or true) information representations. In particular, this logic is most confusing when the results find (as I mentioned in point 1) that alignment to the class average is a *better* predictor than alignment to the experts. This seems to call for some consideration of an alternate explanation, e.g., maybe the students are paying more attention during lectures and remembering more lecture details (as has been shown with previous eeg studies mentioned in the discussion). In fact, the finding that the hippocampus and the “control regions” of primary visual and auditory cortex during the lectures are among the most predictive ROIs seems to support such a conclusion - that this alignment to class is demonstrating a classic subsequent memory effect, which is usually described in terms of rote mnemonic recall rather than conceptual understanding. The fact that the hippocampus shows the strongest effect is consistent with this interpretation as well. In contrast, rather than considering such a conclusion, the authors seem to assume that these patterns represent conceptual understanding per se. Given that alignment-to-experts during the exam seems to be a weaker predictor than alignment to other students during the lectures, do the authors think it is possible that their results reflect rote memorization (maybe due to increased attention) rather than conceptual understanding? Such a result would still be of great importance, and should at least be discussed either way.

We thank the reviewer for coming up with the wonderful “central limit theorem of knowledge” analogy and with your permission, we would like to now incorporate it into the text and also acknowledge you with a footnote -- please let us know if this is permissible. As with the previous comment, we again agree with the reviewer that alignment to class **during lectures** does not necessarily reflect conceptual understanding. We now address this in the discussion (amended text below).

However, it is important to point out that analyses relating same-question alignment and knowledge-structure alignment to performance on **specific exam questions** were conducted within-subjects, and thus the results of these analyses cannot be explained in terms of differences between participants. Furthermore, the exam was built by course staff specifically to assess conceptual understanding and comprised open questions requiring integrative answers, making it difficult to account for students' performance based on subsequent memory effects or rote recall alone. To demonstrate this, we now make the entire set of exam questions available as supplementary text. We thus believe that same-question alignment and knowledge-structure alignment must reflect conceptual understanding of specific topics in individual students.

Discussion (under “Alignment tracks understanding of specific topics”):

“Importantly, exam questions were highly complex and required integrative answers, aimed at assessing conceptual understanding and making it difficult to account for students' performance based on subsequent memory effects or rote recall alone”.

Discussion (under “Learning the right ‘knowledge structure’”):

“Together with our same-question results, the positive correlation between knowledge-structure alignment and exam performance provides key support for our alignment-as-understanding hypothesis. The correlation between alignment-to-class during lectures and exam performance, discussed above, could be explained in terms of coarse-grained individual-difference variables, even if there is no direct link between alignment-to-class and understanding (e.g., conscientious students may show high alignment-to-class during lectures because they closely attend to videos, and obtain good exam scores because they study a lot outside of class, leading to a correlation between alignment-to-class and exam performance). Crucially, the analyses relating same-question alignment and knowledge-structure alignment to performance on specific exam questions were conducted within-subjects, and thus the results of these analyses cannot be explained in terms of differences between participants.”

Supplementary Text

List of exam questions (16 in total).

3. Do the students’ neural data reflect common misconceptions? Given that the best performing student still missed nearly 1/4 of the questions on the final exam (p. 6 and Fig. 2), there is reason to believe that the average understanding of the class does not reflect true understanding of the concepts. The authors address this point by highlighting the (sometimes-observed) correlation between students neural patterns and experts neural patterns. Building on the 2 points mentioned above, the strongest evidence supporting the assertion that expert and student neural alignment predicts learning outcomes is the finding that during one of the analyses of exam questions (but not the other one), alignment-to-experts predicts exam scores (albeit not better than alignment-to-class). The authors interpret these findings on p. 18 as follows: “These findings show that neural alignment of specific question-by-question patterns was associated with better learning outcomes, indicating that concepts that were represented more similarly to the experts (and the class) were the concepts that students better understood.” I am inclined to agree with this interpretation, however, as noted above, focusing on alignment with experts doesn’t seem to tell the whole story (and the rest of the story is interesting!). In fact, despite the fact that much variance is not explained by the correlation between novices and experts (especially during lectures), the authors use these results to suggest that misconceptions shared among students could not be contributing to the shared class representation. I’m not sure their data support such a conclusion. Again, consider the lecture periods, when the alignment-to-class neural data are most strongly predictive of exam performance. During the lectures, Table 2 reports that correlations between alignment-to-class and alignment-to-experts were $r = .28$ on average across all ROIs (or $.24$ without the “control ROIs”). Yet it was neural activity during the lectures (not the exams) that was most predictive of final exam performance (which again, was never even close to 100% perfect for any student). This pattern of results leaves open the possibility that the method presented here could be used

to identify misconceptions, and times when the whole class is misunderstanding a concept presented in the lectures. This would be an incredibly useful finding, and in fact the study of misconceptions has been a major focus of STEM education research for decades. Are there any question items that were missed by the whole class (or most of the class) and for which many students gave a similar incorrect response? If so, then, foiling the authors' logic, neural alignment-to-class should also predict these incorrect answers. Again, such a finding could be a very important diagnostic tool for students and educators alike.

We thank the reviewer for this insight and we agree that it is possible that the method presented here could be used to identify misconceptions. Unfortunately, due to limited power, we could not test this hypothesis in our data. The exam we used included a limited number of questions (16 questions in total, median 46.5 across questions, s.d. 18.0) and we administered it to a relatively small participant pool. As a result, we could not identify enough instances of consistent mistakes to conduct such refined analysis. Nevertheless, the reviewer's point that misconceptions shared among students could be contributing to the shared class representation is well taken. We now discuss this explicitly in the Discussion:

Discussion

"Another caveat is that common misunderstandings would also be reflected in the common signal. These misunderstandings, however, would not be shared by experts, resulting in high alignment across students but low alignment between students and experts."

4. Do response times contribute to the correlation between the two neural measures (alignment-to-class and alignment-to-experts)? In terms of the data mentioned in point 3 (the exam period correlation between students and experts), notably, the authors address a critical alternative explanation for their findings by running a separate set of analyses (reported in the results and supplementary materials) to control for the effects of response time on exam question accuracy. As they put it on p. 19, this analysis "was motivated by the possibility that (i) longer answers might have yielded more stable spatial patterns, and that (ii) response length and quality could be linked (e.g. better answers could be longer)." Their solution was to regress response time out of the exam question scores to provide a measure of "residual scores" that are not correlated with response time. Importantly, they show that regressing out RT from the exam scores does not impact the results in terms of predicting exam scores with neural pattern correlations. This analysis effectively addresses the second point they raised (point ii) that "response length could be linked to response quality". However, because it is correcting the exam scores but not correcting the neural data, it does not seem to fully address the first point (point i) that "longer answers might have yielded more stable spatial patterns". In particular, could RT still have an effect on the correlation between alignment-to-class values and alignment-to-expert values (e.g., the results presented in Fig. 3)? This type of effect is sometimes accounted for by regressing RT from the neural data in the early stages of processing (although it's arguable that interesting neural variance would be lost by such a process). Especially during the self-paced exam questions it seems possible that the length of time spent on a response could affect the neural pattern (in stability or in other ways) and that this could contribute to the correlation between students and experts being stronger during the exam than during the lectures, as well

as being more predictive of exam performance. If RT were found to contribute to these results, such a finding need not invalidate the results, however it could lead to a different interpretation of the alignment between students and experts, especially during the exam questions. What is the authors' take on this question?

The reviewer correctly points out that we did not provide a response length control analysis for the exam results shown in Table 3 and Fig. 3 (right panels). We have now added this control analysis to the manuscript (Table S2). The amended text reads:

Supplementary Results

Correlation between alignment measures during the exam is robust to response length

We examined the link between alignment-to-class and alignment-to-experts during the exam while controlling for response length. While we found a strong positive correlation between these alignment measures across ROIs during both recaps and the exam (Table 3), neural responses during the exam could conceivably be affected by response length as discussed in the main text. To address this, we used a within-participant regression model to predict alignment from answer length. This model yielded a residual error term for each question ("residual score", predicted alignment minus true alignment). We then correlated residual alignment-to-class and residual alignment-to-experts during the exam (Table S2). We found that across ROIs, correlation values were somewhat higher in the control analysis, arguing against a contribution of response length to the effects shown in Table 3.

Region of Interest (ROI)	Corr. Alignment-to-class and alignment-to-experts
	Exam
Angular Gyrus	0.71**
Ant. Cingulate (ACC)	0.88**
Hippocampus	0.49**
Post. Sup. Temporal Gyrus	0.51**
Precuneus	0.71**
Amygdala	0.42**
Early Auditory	0.76**
Early Visual	0.56**

Table S2. Alignment-to-experts is positively correlated with alignment-to-class in response length control. Correlation between alignment-to-class (controlled for response length) and alignment-to-experts (controlled for response length) during the exam is shown. Results are shown in DMN ROIs as well as in control regions in sensory cortex (visual, intracalcarine cortex; auditory, Heschl's gyrus) and in subcortex (amygdala). Green, significant correlation (permutation test, $p < 0.05$, FDR corrected across ROIs). * $p < 0.05$, ** $p < 0.01$, n.s., not significant.

Minor points:

- The “experts” were not experts in the traditional sense of the expertise literature (e.g., professors or practicing professionals), but rather they were more advanced students than the novice learners. Given this point, further description of their mastery of the concepts would be useful. The methods section states that 6 items out of 64 were excluded because the experts’ responses scored below 2. Given that alignment to the experts was a weaker correlate of exam performance (i.e., concept understanding) than other measures, it would be worth a comment about the possible differences between relying on these advanced students versus actual experts in the field. In terms of whether or not these graduate students displayed mastery of the target concepts, reporting a bit more detail about their scores (at least mean and sd of the experts’ scores) would be useful.

We thank the reviewer for this comment. Indeed, the structure of knowledge is likely to change as a function of expertise level, and we should not expect graduate students, postdocs and professors to possess the same structure of knowledge. In this study we sampled two distanced points on this continuous scale: students with little to zero prior knowledge in computer science (zero score in the placement exam) and advanced students with substantive knowledge in CS (85%+ score in the placement exam). We believe that the contrast between these two groups is sufficient for the aims of this study. Nevertheless, we hope to be able to study more subtle differences across levels of expertise in the future. As suggested, we now include more information about the experts and their performance:

- a. Under participants and stimuli: “Experts all had an undergraduate or graduate degree in computer science and reported having significant programming experience as well as knowledge of introduction to computer science course material (≥ 6 on a 7-point Likert scale).”
- b. Under experimental procedures: “The median score for experts was 85 out of 100 (range 69-100, s.d. 14.2).”

- In the caption for Figure 4, there is a typo: “For all ROI analysis results, see table 3” - should read, “table 2”

Typo fixed.

- Clarification needed: In the two y-axis labels on Fig. 4C and on one of the plots in Fig. 5B the label reads “Question score (raw)” whereas the other similar plots read “Question score” and not “raw”. Are these actually different or just inconsistently labeled? (I assume the latter.)

Inconsistent labels fixed.

Again, despite the length of this review, I find the overall claims of the paper to be well supported and the main findings to be of great interest to the field. I would be happy to review a revised submission of the manuscript, if requested.

Signed,

David Kraemer

REVIEWERS' COMMENTS

Reviewer #1 (Remarks to the Author):

The authors have thoughtfully addressed all of my concerns.

They have engaged more deeply with relevant work on using similar techniques to assess the neural basis of the development of expertise in mathematics in both the Introduction and Discussion sections. The engagement with this research in the Discussion section is particularly effective in placing the current results in the context of relevant previous work.

The additional analyses that have been incorporated into the supplement, such as the power analysis and analyses using temporal alignment-to-class scores during lectures, have also enriched the manuscript in my view. The other additions to the Discussion section have also strengthened the manuscript, particularly the discussion of the relationship between the alignment-to-class and alignment-to-expert patterns.

The newly added discussion of a “central limit theorem” of knowledge in the revised Discussion section seems like another way of describing the well-established idea of the “wisdom of crowds” – i.e., that if a sufficient number of independent observers’ responses are amassed and aggregated, individual-level biases and errors should ‘cancel out’ and the average should approach the right answer/way of thinking. As such, I would recommend using the phrase “wisdom of crowds” instead of or in addition to “a ‘central limit theorem’ of social knowledge”, and perhaps citing relevant work on the wisdom of crowds, to connect the idea communicated here to prior demonstrations of this concept.

Reviewer #2 (Remarks to the Author):

Th authors have adequately addressed all of my comments. I feel that it is now suitable to be published and will be a great contribution to the field.

Reviewer #3 (Remarks to the Author):

The authors have thoroughly addressed the comments that I and the other reviewers raised. I think their responses are convincing and the manuscript is improved for these revisions. Therefore I'm happy to recommend this revised article for publication.

Also, the authors graciously requested my permission for the use of the "central limit theorem of

knowledge" analogy, to which I'll happily grant—it's theirs! To me it's a useful way to conceptualize how the wisdom of the crowd may be a useful indicator of the truth, even when the crowd consists of non-experts.

Point-by-point response to the reviewers' comments

Reviewer #1 (Remarks to the Author):

The authors have thoughtfully addressed all of my concerns.

They have engaged more deeply with relevant work on using similar techniques to assess the neural basis of the development of expertise in mathematics in both the Introduction and Discussion sections. The engagement with this research in the Discussion section is particularly effective in placing the current results in the context of relevant previous work.

The additional analyses that have been incorporated into the supplement, such as the power analysis and analyses using temporal alignment-to-class scores during lectures, have also enriched the manuscript in my view. The other additions to the Discussion section have also strengthened the manuscript, particularly the discussion of the relationship between the alignment-to-class and alignment-to-expert patterns.

We thank the reviewer for their contribution to the manuscript and their positive feedback regarding the additional analyses.

The newly added discussion of a “central limit theorem” of knowledge in the revised Discussion section seems like another way of describing the well-established idea of the “wisdom of crowds” – i.e., that if a sufficient number of independent observers' responses are amassed and aggregated, individual-level biases and errors should ‘cancel out’ and the average should approach the right answer/way of thinking. As such, I would recommend using the phrase “wisdom of crowds” instead of or in addition to “a ‘central limit theorem’ of social knowledge”, and perhaps citing relevant work on the wisdom of crowds, to connect the idea communicated here to prior demonstrations of this concept.

We agree with the reviewer's comment. As suggested, we added the following sentence under “Class patterns reflect expert patterns” in the Discussion:

Similar ideas have been conceptualized as the “wisdom of crowds”.

Reviewer #2 (Remarks to the Author):

The authors have adequately addressed all of my comments. I feel that it is now suitable to be published and will be a great contribution to the field.

We thank the reviewer for their valuable contribution and positive feedback.

Reviewer #3 (Remarks to the Author):

The authors have thoroughly addressed the comments that I and the other reviewers raised. I think their responses are convincing and the manuscript is improved for these revisions. Therefore I'm happy to recommend this revised article for publication.

Also, the authors graciously requested my permission for the use of the "central limit theorem of knowledge" analogy, to which I'll happily grant—it's theirs! To me it's a useful way to conceptualize how the wisdom of the crowd may be a useful indicator of the truth, even when the crowd consists of non-experts.

We thank the reviewer for their constructive feedback and for the generous permission to use their analogy.